



# The new BELUGA setup for collocated turbulence and radiation measurements using a tethered balloon: First applications in the cloudy Arctic boundary layer

Ulrike Egerer[1], Matthias Gottschalk[2], Holger Siebert[1], André Ehrlich[2], and Manfred Wendisch[2]

[1]Leibniz Institute for Tropospheric Research, Permoserstr. 15, 04318 Leipzig, Germany
[2]University of Leipzig, Institute for Meteorology, Stephanstr. 3, 04103 Leipzig, Germany

**Correspondence:** Ulrike Egerer (egerer@tropos.de)

**Abstract.** The new BELUGA (Balloon-bornE moduLar Utility for profilinG the lower Atmosphere) tethered balloon system is introduced. It combines a set of instruments to measure turbulent and radiative parameters and energy fluxes. BELUGA enables collocated measurements either at a constant altitude or as vertical profiles up to 1.5 km height. In particular, the instrument payload of BELUGA comprises three modular instrument packages for high resolution meteorological, wind vector and broadband radiation measurements. The collocated data acquisition allows to estimate the driving parameters of the energy balance in various altitudes. Heating rates and net irradiances can be related to turbulent fluxes and local turbulence parameters such as dissipation rates. In this paper the technical setup, the instrument performance, and the measurement strategy of BELUGA are explained. Furthermore, the high vertical resolution due to the slow ascent speed is highlighted as a major advantage of tethered balloon-borne observations. Three illustrative case studies of the first application of BELUGA in the Arctic atmospheric boundary layer are presented. As a first example, measurements of a single layer stratocumulus are discussed. They show a pronounced cloud top radiative cooling of up to $6\,\mathrm{K\,h^{-1}}$. To put this into context, a second case elaborates respective measurements with BELUGA in a cloudless situation. In a third example, a multi layer stratocumulus was probed, revealing reduced turbulence and negligible cloud top radiative cooling for the lower cloud layer. In all three cases the net radiative fluxes are much higher than turbulent fluxes. Altogether, BELUGA has proven its robust performance in cloudy conditions of the Arctic atmospheric boundary layer.

## 1 Introduction

The effects of global warming are most pronounced in the Arctic. The increased warming of the Arctic is often called Arctic amplification, although it includes in a more general sense all processes that are amplified under the special Arctic conditions. For example, besides near-surface air temperature, Arctic amplification is also associated with sea ice loss, which currently takes place at an unanticipated pace (Richter-Menge et al., 2018). However, the atmospheric and surface processes involved in Arctic amplification and their interactions are not fully understood, causing major uncertainties in model projections of future Arctic climate development. Several comprehensive field campaigns in the Arctic have been carried out (Curry et al., 2000;





Uttal et al., 2002; Tjernström et al., 2014; Granskog et al., 2018; Wendisch et al., 2018), but detailed observations of local processes in the transitioning Arctic climate system are still rare.

One of the key players in Arctic amplification are clouds. In particular, persistent low-level clouds are frequently observed in the Arctic atmospheric boundary layer (ABL), influencing the thermal stratification, atmospheric radiation, and as a con-
sequence the entire energy budget (e.g., Sedlar et al., 2011; Shupe et al., 2011). Clouds modify the outgoing and incoming solar and terrestrial radiation and affect the vertical energy transport and turbulent mixing (Brooks et al., 2017). This feeds back on the clouds, making the cloud-radiation-turbulence interactions an intertwined and complicated system. This complexity increases if the clouds are thermodynamically decoupled from lower atmospheric levels and the surface. In this case, the cloud evolution does not necessarily depend on surface energy fluxes (Curry, 1986; Shupe et al., 2013). Additionally, cloud
properties can be significantly influenced by long-range transport of moisture, heat, and aerosol particles (Pithan et al., 2018).

The majority of Arctic clouds is located within the ABL. Shupe et al. (2013) showed that the turbulent boundary layer structure differs significantly for Arctic single and multi layer clouds, indicating that the reduced cloud top radiative cooling in the multi layer case affects the turbulent fluxes within the lower cloud layer. In most climate models, turbulent and radiative fluxes in these low altitudes are underrepresented, which contributes to the model uncertainties (Vihma et al., 2014). For
example, Lüpkes et al. (2010) showed significant discrepancies between observed temperature and humidity profiles compared to re-analysis data in the lowest few hundred meters. However, their analysis is based on cloudless cases; low-level clouds are supposed to significantly increase the discrepancies. To improve climate model parameterizations of surface heat fluxes and cloud properties, Birch et al. (2009) emphasized the importance of in situ observations for all components of the surface energy budget.

Much of the current knowledge is based on observations carried out using research aircraft (e.g., Curry et al., 1988; Tetzlaff et al., 2015) and ground-based remote sensing observations (e.g., Sedlar and Shupe, 2014). However, in particular the lowermost levels, including fog or low-level clouds, are difficult to probe with manned aircraft. These as well as unmanned aerial systems (e.g., Bates et al., 2013; Jonassen et al., 2015) are mainly limited to cloudless situations due to problems in icing conditions. Ground-based remote sensing measurements were used to analyze the interactions between atmospheric radiation
and turbulence in Arctic mixed-phase clouds (e.g., Sedlar and Shupe, 2014), providing simulation-based estimates of the cloud top cooling (Turner et al., 2018). These studies are limited to a vertical resolution of typically 45 m. Further, only very few in situ observations of cloud top entrainment, the evolution of coupled to decoupled clouds (Shupe et al., 2012), and vertical profiles of solar and terrestrial radiation (Becker et al., 2018) exist in the Arctic.

Tethered balloon measurements enable to bridge the gap between surface based and aircraft measurements by probing the
whole vertical profile of the ABL. They have been deployed successfully in the Arctic, for example during the Arctic Summer Cloud Ocean Study (ASCOS; Shupe et al., 2012; Kupiszewski et al., 2013) and in Ny-Ålesund (Lawson et al., 2011; Sikand et al., 2013). Tethered balloons are less affected by icing, and the slow ascent rate enables a high vertical resolution. Depending on the size of the applied balloon, they can lift payloads from a few kilograms up to several tens of kilograms to an altitude of 1 km or more with an endurance of up to several hours. Duda et al. (1991) and Becker et al. (2018) used a tethered balloon
to measure vertical profiles of irradiances and derived heating rates from the measurements. Canut et al. (2016) showed that





it is possible to measure the three-dimensional wind vector using a tethered balloon, whose motion is directly affected by the turbulent wind field. However, so far no combined balloon-borne vertical profile measurements of turbulent and radiative energy fluxes have been reported, although the combined analysis of radiative and turbulent processes is key for understanding the role of clouds in the context of Arctic amplification.

In this study, we introduce the new Balloon-bornE moduLar Utility for profilinG the lower Atmosphere (BELUGA) for collocated in situ measurements of turbulence and broadband solar and terrestrial radiation. BELUGA with its three modular instrument packages was first deployed during the ship based Arctic field campaign Physical feedbacks of Arctic planetary boundary level Sea ice, Cloud and AerosoL (PASCAL), which is part of the extensive observational effort aiming at understanding cloud processes related to Arctic amplification (Wendisch et al., 2018). In the framework of the PASCAL campaign,

the RV *Polarstern* accessed the sea-ice covered area north of Svalbard in early summer 2017, and drifted during a two-week period attached to an ice floe. PASCAL included substantial instrumentation of different research groups (Macke and Flores, 2018; Wendisch et al., 2018) and is associated with the concurrent aircraft campaign Arctic CLoud Observations Using airborne measurements during polar Day (ACLOUD).

The present study demonstrates the possibility to concurrently measure vertical profiles of turbulence and radiation parame-
ters including energy fluxes using a tethered balloon in the cloudy Arctic ABL. The technical specifications and data analysis of the new setup are introduced in Sect. 2 to Sect. 4. The results of measurements with BELUGA in three typical weather situations (single layer cloud, cloudless conditions, multi layer clouds) observed during PASCAL are presented to illustrate the potential of the new setup (Sect. 5). A summary and a discussion of limitations and potential for future employment of the balloon system are given in Sect. 6.

## 2   Observational

### 2.1   Combined instrument setup

A helium-filled tethered balloon (Fig. 1a) with a volume of $90\,\mathrm{m^3}$ has been deployed with a scientific payload up to $10\,\mathrm{kg}$. The balloon operates in altitudes between ground and $1500\,\mathrm{m}$ at maximum wind speeds of $15\,\mathrm{m\,s^{-1}}$. An electric winch retains the balloon with climb and descent rates of typically $1$–$3\,\mathrm{m\,s^{-1}}$. The balloon is captured by a $3\,\mathrm{mm}$ thick Dyneema® tether with
$900\,\mathrm{daN}$ strength. It is equipped with an emergency deflation system in case of tether failure. The tethered balloon system can operate inside clouds and at light icing conditions.

For operation on the tethered balloon, three instrument packages were developed: an ultrasonic anemometer package (UP, Fig. 1b), a hot-wire anemometer package (HP, Fig. 1c) and a payload measuring solar and terrestrial broadband irradiances (BP, Fig. 1d). The packages can be deployed on the balloon considering three main configurations of turbulence and radiation
measurements (Fig. 2): Configuration 1 is designed for combined turbulence and radiation measurements in rather low wind conditions up to $10\,\mathrm{m\,s^{-1}}$, when the lift of the balloon allows a larger mass of the payload. It includes the UP for turbulence and one BP for radiation measurements. For strong wind conditions, the payload is reduced to reach a sufficient maximum altitude. This configuration 2 comprises the HP and one BP. Configuration 3 is applied in most favorable conditions (low and



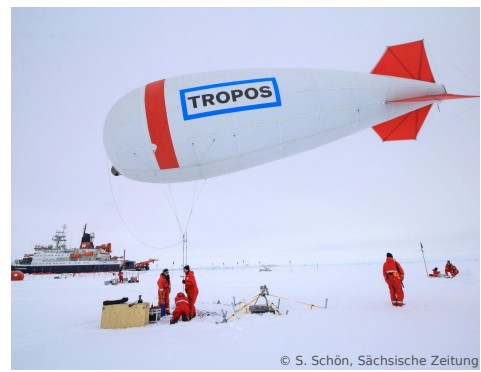

(a) Tethered balloon system

© S. Schön, Sächsische Zeitung

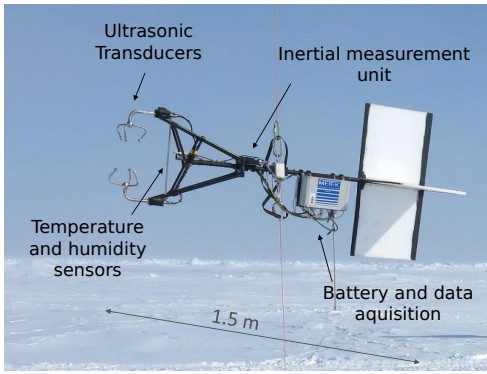

(b) Ultrasonic anemometer package (UP)

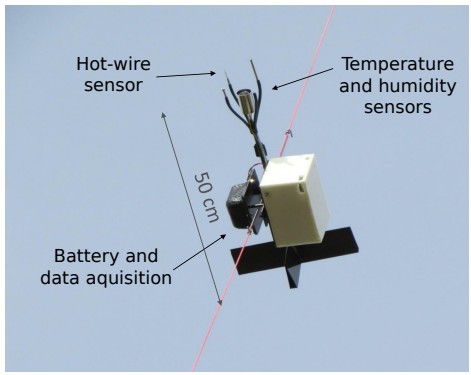

(c) Hotwire anemometer package (HP)

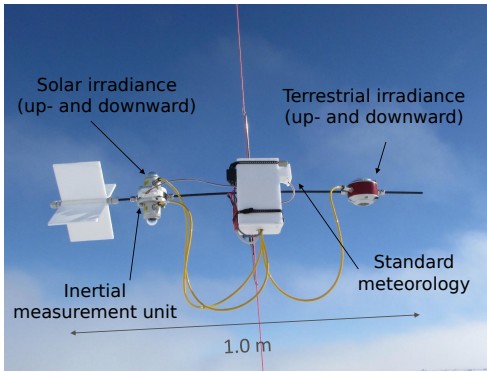

(d) Broadband radiation package (BP)

**Figure 1.** The tethered balloon system on the ice floe next to RV *Polarstern* (a) and photographs of the individual instrument packages for turbulence (b and c) and radiation (d).

uniform wind speed), comprising of the UP and two BPs. A separate standard meteorology package routinely measures air temperature, relative humidity, wind speed and altitude, and transmits the data to ground for online monitoring. The ultrasonic anemometer, with a weight of around $6 \, \mathrm{kg}$, is attached at a fixed point in a distance of $20 \, \mathrm{m}$ below the balloon; the other more lightweight payloads HW and BP ($< 3.5 \, \mathrm{kg}$) can be flexibly attached to the tether at any distance from the balloon. The modular

5   instrument set-up allows to adjust instruments to different scientific questions.

All instrument packages are equipped with basic meteorological measurements. For all instrument packages, the barometric altitude is calculated from the barometric pressure $p_{\mathrm{b}}$ by

$$z_{\mathrm{b}} = \frac{T_0}{L_0} \cdot \left( 1 - \frac{p_{\mathrm{b}}}{p_0}^{R \cdot L_0 / g} \right) \tag{1}$$

with the standard adiabatic lapse rate $L_0 = 6.5 \, \mathrm{K \, km^{-1}}$ and the gas constant for dry air $R = 287 \, \mathrm{J \, kg \, K^{-1}}$. The ground

10   temperature $T_0$ is measured at a nearby meteorological mast. The ground pressure $p_0$ is a 10 second average of the payload's



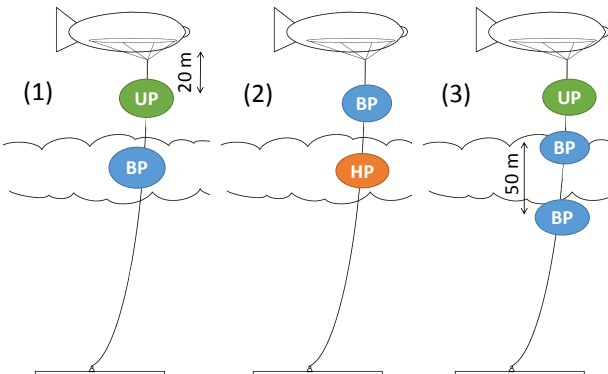

**Figure 2.** Three main instrument configurations on the tethered balloon: (1) configuration 1 with an ultrasonic anemometer package (UP) and a broadband radiation package (BP), (2) configuration 2 with a BP and a hot-wire anemometer package (HW) and (3) configuration 3 with an UP and two BPs. Due to the modular approach, further configurations are possible. Distances and dimensions are not to scale.

$p_b$ before the start of the flight. The barometric pressure is corrected for the pressure change over each flight. This standardized procedure ensures comparable altitudes.

## 2.2 Measurement strategy

The sampling strategy is based on two different approaches: (1) Keeping BELUGA at a constant altitude for a time period of typically 10–15 minutes. In this case, the data provide a statistical basis for turbulent flux estimates or to characterize the time evolution of the radiative cloud properties. (2) A continuous ascent or descent through the ABL yields a vertical profile to study the vertical distribution of ABL parameters.

Measurements close to the ground are used for a comparison to the 10 m high meteorological mast. Figure 3 shows an exemplary height record for one complete flight, which includes all elements of the measurement strategy. After an hour-long measurement period at ground, a continuous ascent is performed. This provides an overview of the boundary layer using the online measurements of the meteorology instrument package and is the basis for the measurements of the second part of the flight: Levels for continuous measurements at fixed altitudes were defined around the temperature inversion and inside the cloud layer.

## 3 Instrument packages

### 3.1 Ultrasonic anemometer package (UP)

The ultrasonic anemometer package (UP, Fig. 1b) includes wind vector $\boldsymbol{u_S}$ and sonic (virtual) temperature $T_v$ measurements with a sampling frequency of $f_s = 50$ Hz, accompanied by a thermometer and a capacitive humidity sensor. Ultrasonic anemometer measurements provide the three-dimensional wind vector and are considered to be unaffected by cloud droplets





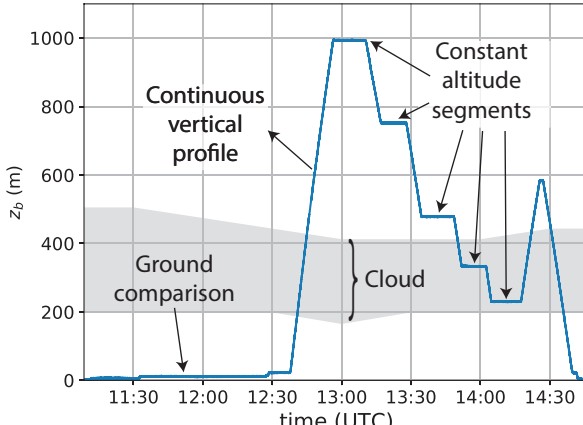

**Figure 3.** Measurement strategy illustrated by the balloon altitude time series of 5 June 2017: The two approaches are (1) constant altitude segments and (2) a continuous vertical profile. The shaded area illustrates the vertical extent of the cloud layer, which is determined from Cloudnet data (Griesche et al., 2019).

(Siebert and Teichmann, 2000). To measure the geo-referenced wind vector from a moving platform such as a tethered balloon, the attitude and motion of the instrument has to be measured precisely and the wind vector measured in the sonic frame has to be corrected for this motion (Canut et al., 2016). The attachment point 20 m below the balloon allows for rotation around the tether and pitch alignment. A wind vane ensures that the anemometer turns towards the mean wind direction. The UP is complemented by a power supply based on lithium-polymer batteries with 6.4 Ah at 12.8 V. This allows for an operation of four hours in Arctic summer conditions. The data acquisition consists of a serial data logger recording on an SD card. With all sensors, data acquisition and surroundings the total mass of the UP adds up to 6 kg.

### 3.1.1 Wind vector

The wind measurements are performed using a Metek uSonic-3 anemometer, complemented by a GPS-assisted inertial measurement unit (IMU, Table 1) for the attitude and motion correction. A data logger collects the two data streams from the sonic and the IMU via two serial ports. The data streams are synchronized by help of an analog pulse per second (PPS) signal, which is sent from the IMU to the sonic data acquisition. A careful temporal synchronization of sonic and IMU measurements is a basic requirement for a successful transformation of the sonic wind vector. The sonic transducers are heated for anti-icing. The IMU's accuracy for roll and pitch angles is $0.1°$ for angles not exceeding $\pm 10°$. The sonic anemometer has a resolution of $1\,\mathrm{cm\,s^{-1}}$ and 10 mK, respectively. Further instrument characteristics are summarized in Table 1.

Transforming the wind vector, as measured in the sonic framework $\boldsymbol{u_S} = (u_S, v_S, w_S)$, into an earth-fixed reference system is a standard procedure for research aircraft (Wendisch and Brenguier, 2013; Lenschow, 1986). In contrast, for a tethered balloon it is more challenging, because the turbulence to be measured drives the motion of the balloon within the same fre-





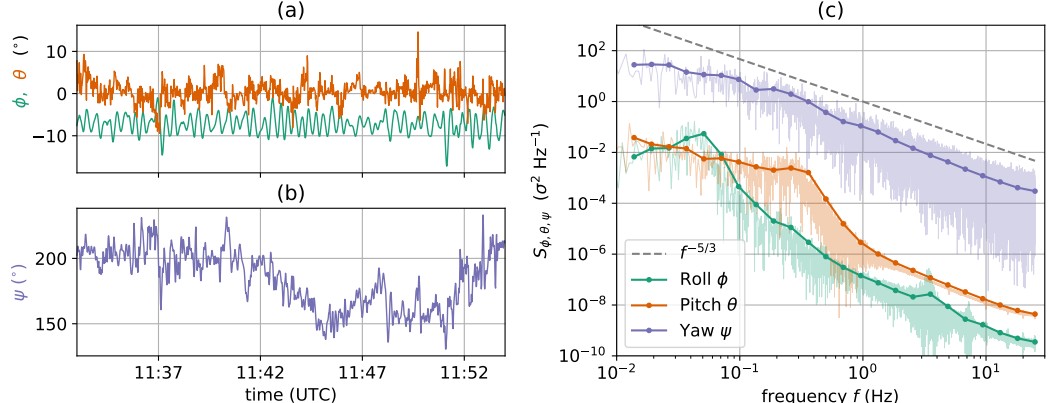

**Figure 4.** Time series (a and b) and power spectral density $S$ (c) of the Euler angles roll $\phi$, pitch $\theta$ and yaw $\psi$ measured by the IMU on the UP. The time series used for the spectra calculations was recorded on 5 June 2017, 11:33–11:54 UTC, with the balloon kept at 20 m altitude.

quency range. This instrument motion $\boldsymbol{u}_{\mathrm{IMU}} = (u_{\mathrm{IMU}}, v_{\mathrm{IMU}}, w_{\mathrm{IMU}})$ is measured by the IMU. The geo-referenced wind vector $\boldsymbol{u_E} = (u_E, v_E, w_E)$ is obtained by applying the following transformation:

$$\boldsymbol{u}_E = \boldsymbol{T} \cdot (\boldsymbol{u}_\mathrm{S} + \boldsymbol{\Omega} \times \boldsymbol{L}) + \boldsymbol{u}_{\mathrm{IMU}}, \tag{2}$$

with the rotational velocities $\boldsymbol{\Omega}$ as measured by the IMU and the lever arm vector $\boldsymbol{L} = (59\,\mathrm{cm}, 0, 0)$ between IMU and sonic.
The transformation matrix $\boldsymbol{T}$ is a function of the Euler angles roll $\phi$ , pitch $\theta$ and yaw $\psi$ , which are provided by the IMU.

Figure 4 illustrates the instrument motion in a 20 min time series of the Euler angles and the corresponding frequency spectra. The smoothed spectra result from averaging over logarithmic equidistant bins. The yaw angle spectrum is not characterized by a single spectral peak but shows almost a $-5/3$ slope indicating inertial subrange scaling (Wyngaard, 2010). This behavior is due to the fact that the probe can freely rotate about its vertical axis and the yaw angle variations are dominated by the turbulent
wind direction rather than the balloon motion. This is different for the roll and pitch angles, showing distinct peaks at 0.05 and 3 Hz for the roll angle and 0.3 Hz for the pitch angle. The roll angle peak is most probably a result of lateral motion of the balloon itself, whereas the pitch angle might be a combination of balloon motion and a pitch motion of the payload. In the time series, the roll angle oscillation is obvious.

The result of a geo-referenced wind vector measurement from a moving platform is affected by misalignment between the
IMU and the wind sensor. Small offset angles in roll and pitch can be estimated by applying post-processing algorithms. Here, we apply a correction procedure based on Wilczak et al. (2001), who suggest that for a sufficiently long record the mean vertical wind vanishes. The simplified transformation equations based on Eq. 2 yield the misalignment angle offsets. For the presented study, $\Delta\theta = 0.8°$ for pitch and $\Delta\phi = 5.15°$ for roll offset are determined in a defined time period of roughly 30 minutes, where the balloon was kept near the ground. Those angle offsets are applied to all campaign data by adding to the measured angles in
$\boldsymbol{T}$. The resulting geo-referenced wind speed components serve to calculate the wind direction: $d_E = \arctan\left(\overline{v}_E/\overline{u}_E\right)$.





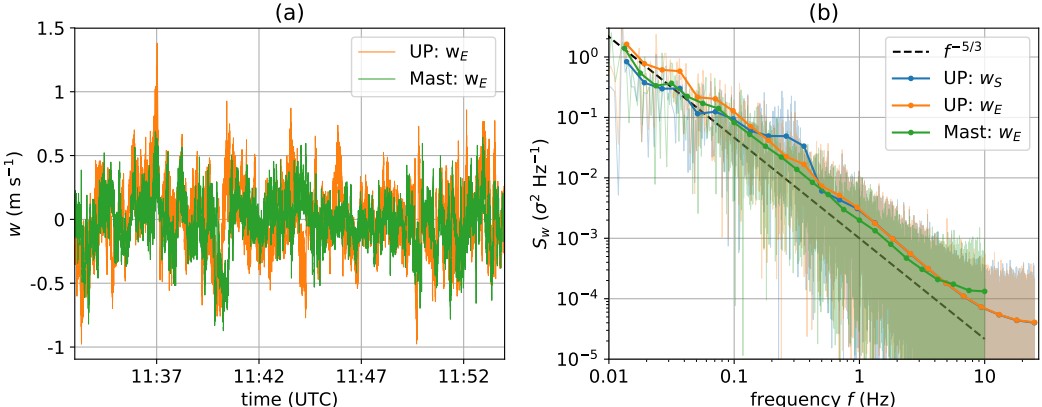

**Figure 5.** Time series of the vertical wind speed $w$ (a) and power spectral density $S_w$ (b). Wind speeds are shown for the UP measurements with (orange) and without motion correction (blue, only plotted in the right panel). As reference, the sonic data measured on the mast in a height of $10\,\mathrm{m}$ are presented (green). The time period is the same as in Fig. 4, the mean horizontal wind speed for this period was $1.7\,\mathrm{m\,s^{-1}}$.

The results of the wind vector calculation are verified by comparing a $20\,\mathrm{min}$ data set recorded in around $20\,\mathrm{m}$ altitude to measurements from a $10\,\mathrm{m}$ meteorological mast situated close to the balloon site. On the mast, an ultrasonic anemometer measures the wind speed with $20\,\mathrm{Hz}$ sampling frequency. Instrument characteristics are listed in Table 1. Figure 5 shows a time series of vertical wind speed measured by the sonic anemometer installed on the mast and the balloon payload UP, as

well as the power spectral densities. The measured vertical wind speed fluctuates around values close to zero and the standard deviation on the balloon sonic $\sigma_w = 0.19\,\mathrm{m\,s^{-1}}$ is close to the mast sonic $\sigma_w = 0.16\,\mathrm{m\,s^{-1}}$. In the spectra shown in Fig. 5b, both curves reveal an inertial subrange with equal power spectral density values. The curves flatten at frequencies above $5\,\mathrm{Hz}$ for the mast and $10\,\mathrm{Hz}$ for the balloon, respectively. This flattening is due to uncorrelated noise. With a spectral noise floor (indicated by $^{(n)}$) of $S_w^{(n)} \approx 4\cdot10^{-5}\,\sigma^2\,\mathrm{Hz^{-1}}$ and the Nyquist frequency of $f_{\mathrm{Ny}} = f_s/2 = 25\,\mathrm{Hz}$, the standard deviation due to

uncorrelated noise

$$\sigma_w^{(n)} = \sqrt{S_w^{(n)} \cdot f_{\mathrm{Ny}}} \approx 0.03\,\mathrm{m\,s^{-1}} \tag{3}$$

can be estimated for the balloon UP, and $\sigma_w^{(n)} \approx 0.04\,\mathrm{m\,s^{-1}}$ for the mast. The noise level for the UP horizontal wind speed is five times higher than for vertical wind speed (not shown here). The standard deviation due to uncorrelated noise for the sonic temperature is $\sigma_{T_v}^{(n)} \approx 0.05\,\mathrm{K}$.

In addition to the spectrum of the motion-corrected vertical velocity component (orange curve), the spectrum for the uncorrected vertical sonic component (blue curve) are shown in Fig. 5b. The blue curve clearly shows the spectral peak around $0.3\,\mathrm{Hz}$ caused by the pitch motion (compare Fig. 4c). This motion is not visible anymore in the orange curve indicating a successful correction procedure and a clear power law. However, under certain conditions – in particular near ground with increased wind fluctuations due to convection – the balloon motion cannot be completely eliminated from the sonic measurements. This is





probably due to the fact that the wind fluctuations which drive the balloon motion are in the same frequency range as the turbulence to be measured.

### 3.1.2 Temperature

The UP includes a PT100 thermometer as reference and a fast-response thermocouple sensor for temperature measurements.
The temperature readings are sampled with 16 bit resolution and 50 Hz sampling frequency by additional analog inputs of the sonic anemometer. Therefore, temperature and wind velocity data are synchronized. The PT100 provides a slow-response, but relatively accurate temperature measurement. The thermocouple sensor is calibrated for each flight seperately to the PT100 sensor via a linear regression. For both the thermocouple and the PT100 sensor, the time-response error is corrected similar to McCarthy (1973) by the following relation:

$$T_\tau = T + \frac{\partial \widetilde{T}}{\partial t} \cdot \tau. \tag{4}$$

The tilde represents a Savitzky-Golay filtered temperature signal. The $e^{-1}$ time constants $\tau_{\text{PT100}} \approx 10$ s and $\tau_{\text{thermo}} \approx 0.64$ s are determined for a rapid temperature change when descending through a sharp inversion layer, simulating a first-order step response experiment. The virtual temperature is directly measured by the sonic anemometer with a time response below 20 ms and therefore serves as a reference. Typically, sonic temperature measurements show an offset due to the measurement princi-
ple. This offset is corrected by a calibration with the virtual temperature calculated from temperature and humidity sensors.

In Fig. 6, the PT100 and thermocouple temperatures are compared to the temperature calculated from the sonic virtual temperature by $T_{\text{sonic}} = T_{\text{v}} \cdot (1 + 0.61 \cdot q)^{-1}$. The thermocouple data resolve small features inside the temperature inversion layer, which are also seen in the sonic temperature. In the time-response corrected PT100 signal, details and fluctuations are not completely resolved. The comparison with the radiosounding (Vaisala RS92-SGP, Schmithüsen, 2017) illustrates the
capability of our measurements to resolve small scale structures. The temperature difference between balloon and radiosonde measurements is probably caused by the two hour time lag between the measurements.

### 3.1.3 Relative humidity

Relative humidity (RH) is measured with an EE08 capacitive humidity sensor. Similar to the temperature sensors, the relative humidity signal of the UP is time-response corrected with $\tau \approx 6$ s, as specified by the manufacturer. The correction for sensor
inertia improves the resolution of vertical structure in RH, but still the sensor is slow compared to the temperature measurements. A weak point of the capacitive humidity sensor is its deficiency to reproduce values close to saturation in clouds. In contrast, the applied radiosonde type (Schmithüsen, 2017) shows small RH measurement errors being systematically below 4 % in the high-latitude troposphere (Ingleby, 2017). Therefore, the EE08 sensor is calibrated by comparing to the temporally closest radiosonde. With the applied corrections, the measured humidity inside clouds still varies between 95 % and 100 %.




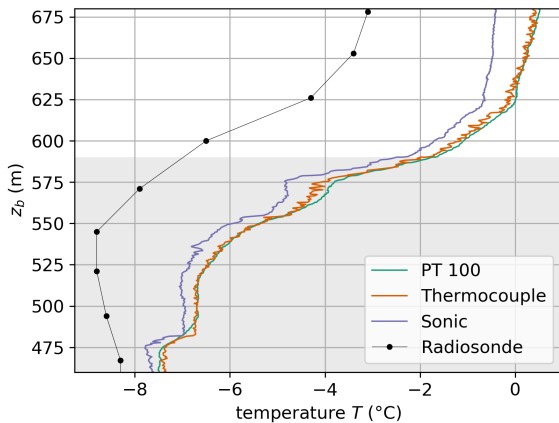

**Figure 6.** Comparison of balloon-borne temperature measurements measured with the UP descending at 10:00 UTC on 7 June 2017. The plot shows temperature $T$ as a function of barometric altitude $z_b$ around a temperature inversion capping the cloud layer (shaded area, determined from relative humidity and Cloudnet data). The temperature vertical profile includes the time-response corrected PT100, corrected and calibrated thermocouple data and the temperature calculated from the sonic temperature. Plotted data are averaged over 0.5 s intervals. The temperature profile as observed from the radiosonde about 2 hours later is shown for comparison.

## 3.2 Hot-wire anemometer package (HP)

The hot-wire anemometer package (HP, Fig. 1c) is used to measure the one-dimensional wind speed with a temporal resolution of 111 Hz. The centerpiece is a 5 μm hot-wire sensor connected to a small constant temperature anemometer circuit (Dantec MiniCTA). The electrical circuit keeps the temperature of the wire sensor constant, the voltage output of the circuit is related to the wind speed. For reference, the dynamic pressure is measured by a Pitot-static probe, connected to a differential pressure gauge. The hot-wire and the Pitot-static sensor face the mean airflow by means of a wind vane. Icing of the Pitot-static probe was observed only for one flight during the entire campaign.

The hot-wire sensor is calibrated for each flight against the wind speed derived from the Pitot-static probe

$$u_{\mathrm{p}} = \sqrt{\frac{2\rho}{p_{\mathrm{dyn}}}} \tag{5}$$

by means of a fourth-order polynomial regression (Jørgensen, 2005). The dynamic pressure $p_{\mathrm{dyn}}$ measured by the Pitot-static probe has been calibrated in a wind tunnel against a highly accurate differential pressure gauge. The individual calibration for each flight is necessary to overcome the temperature dependence of the hot-wire reading. For intercomparison, simultaneous measurements of the HP and the UP are analyzed in Fig. 7, showing a vertical profile on 8 June 2017. Wind speed measurements of all instruments agree well (Pearson correlation coefficient $R^2 = 0.94$ between hot-wire and sonic), as illustrated in Fig. 7a. Figure 7b compares energy spectra of hot-wire wind speed for three constant altitude segments. The hot-wire is able to resolve turbulent structures up to the Nyquist frequency $f_{\mathrm{Ny}} = 55$ Hz. The spectra are averaged over logarithmic equidistant bins



**Table 1.** Characteristics of the main instruments included in the particular balloon-borne sensor packages and on the 10m meteorological mast.

| Instrument | Manufacturer | Measured quantity | Range | Temporal resolution |
|---|---|---|---|---|
| TURBULENCE I: ULTRASONIC ANEMOMETER PACKAGE (UP) | | | | |
| Ultrasonic anemometer uSonic-3 Class A | METEK GmbH, Germany | $\boldsymbol{u}_\mathrm{S}, T_v$ | $0$–$20\,\mathrm{m\,s^{-1}}$ | 50 Hz |
| Inertial Measurement Unit iμVRU+iTILT | iMAR Navigation GmbH, Germany | $\boldsymbol{u}_\mathrm{IMU}, \boldsymbol{\Omega},$ $\phi, \theta, \psi, p_\mathrm{b}$ | $0$–$100\,\mathrm{m\,s^{-1}}$, $250°\,\mathrm{s^{-1}}$ | 50 Hz |
| Thermocouple 5TC-TT-KI-10-1M | OMEGA Engineering, Inc., USA | | | |
| Transducer LKM 212 for thermocouple | LKM electronic GmbH, Germany | $T$ | $-20$–$80°\mathrm{C}$ | 10 Hz |
| PT 100 with LKM 467 | LKM electronic GmbH, Germany | $T$ | $-20$–$80°\mathrm{C}$ | 1 Hz |
| Humidity sensor EE08 | E + E Elektronik GmbH, Austria | RH | $0$–$100\,\%$ | 1 Hz |
| TURBULENCE II: HOT-WIRE ANEMOMETER PACKAGE (HP) | | | | |
| Hot-wire 55UUP with MiniCTA | Dantec Dynamics A/S, Denmark | $u$ | $0$–$50\,\mathrm{m\,s^{-1}}$ | 111 Hz |
| Thermocouple 5TC-TT-KI-10-1M | OMEGA Engineering, Inc., USA | | | |
| Transducer LKM 212 for thermocouple | LKM electronic GmbH, Germany | $T$ | $-20$–$80°\mathrm{C}$ | 10 Hz |
| PT 100 with transducer LKM 467 | LKM electronic GmbH, Germany | $T$ | $-20$–$80°\mathrm{C}$ | 1 Hz |
| Pitot-static probe with differential pressure sensor AMS 5812 0000-D | AMSYS GmbH & Co. KG, Germany | $p_\mathrm{dyn}$ | $0$–$5.17\,\mathrm{hPa}$ | 2 Hz |
| Barometric pressure sensor AMS 5812 0150-B | AMSYS GmbH & Co. KG, Germany | $p_\mathrm{b}$ | $760$–$1200\,\mathrm{hPa}$ | 2 Hz |
| Humidity sensor EE08 | E + E Elektronik GmbH, Austria | RH | $0$–$100\,\%$ | 1 Hz |
| BROADBAND RADIATION PACKAGE (BP) | | | | |
| CMP-3 Pyranometer (up- and downward) | Kipp & Zonen B.V., The Netherlands | $F_\mathrm{net,solar}$ | $0.3$–$2.8\,\mathrm{\mu m}$ | 12 Hz |
| CGR-4 Pyrgeometer (up- and downward) | Kipp & Zonen B.V., The Netherlands | $F_\mathrm{net,terr}$ | $4.5$–$42\,\mathrm{\mu m}$ | 12 Hz |
| Inertial Measurement Unit 10-DOF | Adafruit Industries, LLC, USA | $\phi, \theta, \psi, p_\mathrm{b}, T$ | $250°\,\mathrm{s^{-1}}$ | 10 Hz |
| Meteorology sensor BME280 | Bosch Sensortec GmbH, Germany | $p_\mathrm{b}, T$, RH | $300$–$1100\,\mathrm{hPa}$ | 10 Hz |
| 10 m METEOROLOGICAL MAST | | | | |
| Ultrasonic anemometer USA-1 | METEK GmbH, Germany | $\boldsymbol{u}_\mathrm{S}, T_v$ | $0$–$50\,\mathrm{m\,s^{-1}}$ | 20 Hz |
| Thermo-/ Hygrometer HMHP40 | Vaisala Corporation, Finland | $T$, RH | $-40$–$180°\mathrm{C}$, $0$–$100\,\%$ | 1 Hz |
| Barometer PTB220 | Vaisala Corporation, Finland | $p_\mathrm{b}$ | $500$–$1100\,\mathrm{hPa}$ | 1 Hz |



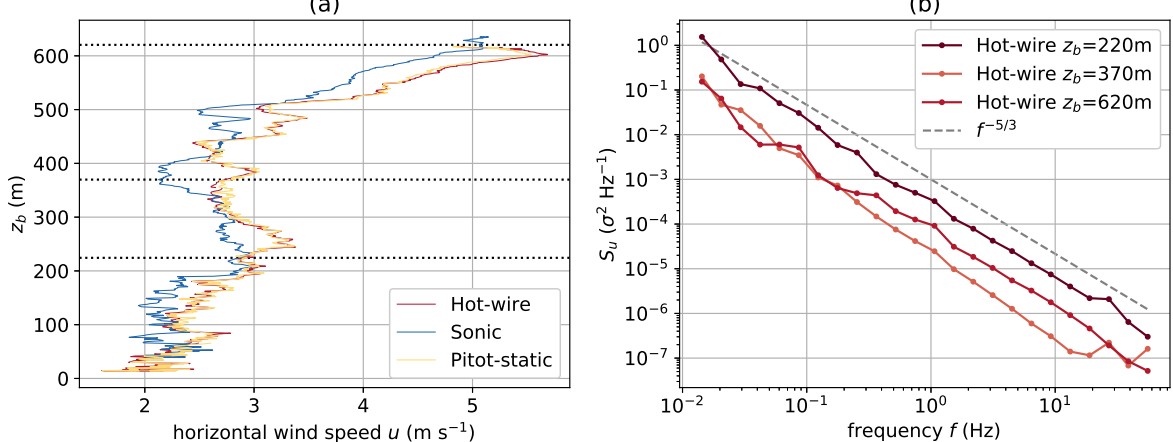

**Figure 7.** Horizontal wind speed $u$ measured on 8 June 2017 with UP and HP simultaneously. Panel (a) compares the vertical profile of wind speed for measurements with the sonic anemometer, hot-wire anemometer and the Pitot-static probe. The HP was attached 25 m below the UP. Panel (b) contrasts power spectral densities for horizontal wind speed $S_u$ measured with the hot-wire anemometer on three constant altitude segments of ~1000 s length (dotted horizontal lines in panel a). The data for the spectra was sampled during an ascent shortly before the descent profile of panel (a).

without applying any anti-aliasing or low-pass filtering, therefore, a slight flatten or even an increase of the spectrum is visible at high frequencies. Further, the hot-wire spectra exhibit some irregularities in the frequency range of the balloon movement (around 0.1 Hz), as the HP instrument motion cannot be compensated. The spectral noise floor of the hot-wire is below $10^{-7}\ \sigma^2\,\mathrm{Hz}^{-1}$. Therefore, a standard deviation due to uncorrelated noise of below $0.2\ \mathrm{cm\,s}^{-1}$ can be estimated.

Siebert et al. (2007) showed that impacting cloud droplets are visible in a hot-wire data set as sharp signal peaks with a duration of ~0.5 ms at a given flow speed of $9\ \mathrm{m\,s}^{-1}$. The general influence of impacting droplets on the hot-wire reading is supposed to be smaller for Arctic clouds compared to the cases shown by Siebert et al. (2007) due the the lower true airspeed for the balloon in combination with lower droplet number concentrations. Further, the peaks include at maximum one data point because the sampling frequency is much smaller than the length of those spikes. The data spikes due to droplet impacts

are eliminated with a simple filter algorithm, removing all single data values exceeding $5\sigma$. Finally, the same filter algorithm is applied to overcome a technical problem with the data acquisition system, when two subsequent data telegrams of the analog signals were mixed.

Thermodynamic measurements with the HP, including applied corrections, are the same as for the UP (Sect. 3.1.2). Both temperature sensors on the HP show a small number of peaks in the data (in the order of 4 s length and 1.5 K magnitude)

on two days inside a cloud, probably due to cloud droplet collisions. For the flight performed on 8 June 2017 (the only flight with simultaneous hot-wire and sonic measurements), the thermocouple data correlate with the sonic virtual temperature





measurements with $R^2 = 0.99$. The correlation of the RH measurement on UP with the HP measurement on the 8 June flight is $R^2 = 0.98$.

The hot-wire package is complemented by a power supply with 5.2 Ah at 14.8 V, allowing operating times of more than five hours. A sensor boom at the tip of the probe ensures undisturbed hot-wire and temperature measurements. A RaspberryPi

single-board computer collects and records the data separately at 111 Hz for the fast sensors via a 24 bit analog digital converter and at 2 Hz for the slow sensors. The slow and the fast data stream are synchronized by GPS time. The total mass of the package is 1.2 kg.

### 3.3 Braodband radiation package (BP)

The broadband radiation package (BP, Fig. 1d) measures the solar and terrestrial upward and downward irradiances. Two

Kipp & Zonen CGR4 pyrgeometers cover the irradiance in the terrestrial spectral range between 4.5–42 µm. The pyrgeometers are calibrated by the manufacturer, referring to the world radiometric reference. The uncertainties for the individual sensors are below 5 % and the e-folding ($e^{-1}$) response time is below 6 s. According to the manufacturer (e.g., Kipp & Zonen B.V.; Becker et al., 2018), the operation temperature is down to -40°C, while the non-linearity of the sensor calibration for changing environmental conditions is below 1 % for temperatures down to -20°C. A new metal instrument body was designed to reduce

weight and to mount both pyrgeometers facing opposite directions (downward and upward). Becker et al. (2018) revised their Kipp & Zonen pyrgeometer in similar fashion and tracked the calibration coefficients before and afterwards, finding no change in sensor characteristics.

Two Kipp & Zonen CMP3 pyranometers provide the solar irradiance in the spectral range between 0.3–2.8 µm. The manufacturer calibrated these instruments referring to the world radiometric reference and determined the uncertainties to 3 %. The

$e^{-1}$ response time ranges around < 6 s, which is similar to the pyrgeometers. The stability of the pyranometer measurements requires stable environmental temperatures to ensure thermal equilibrium of the sensor and the environment. For ascents and descents within the ABL, where the air temperature profile may change within tens of meters, this requirement might not be fulfilled. For a temperature change of 5 K h$^{-1}$ the pyranometer has an offset of 5 W m$^{-2}$. With an average balloon ascent speed of 1 m s$^{-1}$ and temperature inversions of up to 10 K, higher offsets need to be considered. However, assuming that this

temperature change equally influences up- and downward facing sensors, this systematic bias will vanish when calculating the net irradiance. Two pyranometers are mounted on a joint body facing downward and upward to measure the net solar radiation. The sensor body is equipped with an IMU, including a pressure and temperature sensor, to track the attitude and altitude of the instrument package.

Due to weight limitations of the sensor package, no radiation shield or ventilation is mounted on the instrument. This is

justified by a good heat conduction between the housing and the dome of the pyrgeometer (Bucholtz et al., 2010) and by a sufficient air stream during the balloon-borne operations to adjust the housing to the outside temperature. The pairs of pyranometers and pyrgeometers are mounted on a glass fiber rod, which is attached and leveled by a flexible mounting to the balloon tether. A wind vane aligns the instrument upwind and damps movement of the sensors.





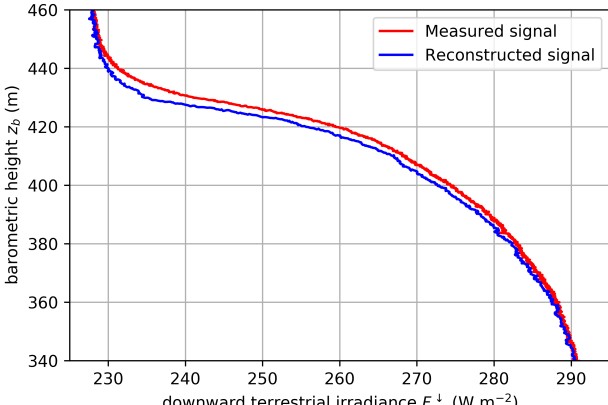

**Figure 8.** Vertical profile of downward terrestrial irradiance measurement on 5 June 2017 (red) in the cloud top region. The blue line displays the reconstructed signal.

Similar to the HP, a RaspberryPi 3b single-board computer records data at 12 Hz sampling frequency for the radiometers and 10 Hz for all other sensors. Batteries supply the instrument with power for up to eight hours. Combined with the housing and the battery, the total mass is 2.2 kg.

### 3.3.1 Reconstruction of high-resolution time series

Solar and terrestrial irradiances change fast in the cloud top region. Both pyranometer and pyrgeometer have a slow response time of < 6 s, which smoothes the irradiance profile and underestimates the drastic change of the net irradiance at cloud top. To correct for these effects, a correction algorithm described by Ehrlich and Wendisch (2015) is applied: The measured time series is deconvoluted by the Fourier transform of the time response assuming that the time response follows an exponential decay. To reduce the impact of electronic noise on the deconvolution, a low pass filter with cut-off frequency of 0.2 Hz is applied. For discontinuities in the irradiance time series, which are expected at cloud top, the deconvolution is limited by the Gibb phenomenon, which describes fluctuation the of the reconstruction around a discontinuity. Applying an additional moving average filter with 1 s window length damps these effects.

Figure 8 shows the downward terrestrial irradiance measured on 5 June 2017, 17:00 UTC, when the BP ascended through the cloud top. The uncorrected irradiance decreases from $290\,\mathrm{W\,m^{-2}}$ to $230\,\mathrm{W\,m^{-2}}$ within 100 m. The reconstructed signal increases stronger and reaches the above-cloud value of $230\,\mathrm{W\,m^{-2}}$ at a lower altitude. This difference can be important, when quantifying the cloud top height via the maximum cloud top cooling in relation to the inversion base height. These altitudes frequently do not coincide in the Arctic due to humidity inversions (Solomon et al., 2011).





### 3.3.2 Ice flagging

Icing of the sensor domes results from water vapor deposition or contact freezing of supercooled droplets in clouds or fog and can significantly influence the measurements of solar and terrestrial irradiance. The sensor icing strongly depends on the environmental conditions such as temperature, liquid water content, cloud droplet size, and horizontal wind speed (Baumert et al., 2018), which makes an automated filtering of the data impossible. To monitor icing of the sensor domes, the BP contains a digital camera pointing towards the pyranometers. The images are visually analyzed for icing. Affected data are flagged and not considered for the data analysis. The image frequency was increased from 1 min to 10 s during the campaign to better resolve the icing periods and gain additional information on cloud top and base height. Icing occurred during most of the flights (60 %) when penetrating a cloud, mainly on the upstream side of the pyranometer dome. Above the cloud, the intense solar radiation could melt the ice within 30 min. The pyrgeometer domes did not show any signs for icing during PASCAL, likely due to the less convex shape of the dome and a more laminar air flow around it.

## 4 Data analysis methods

### 4.1 Derivation of turbulence parameters

The measurement strategy introduced in Sect. 2.2 results in two methods to derive vertical profiles of turbulence parameters: (1) Averaging over data segments at constant altitude yields a limited number of data points on the vertical profile, but a reasonable statistical basis. The vertical resolution depends on the number of constant altitude segments. For the constant altitude segments, turbulent fluctuations are obtained by removing any trend of the data time series. (2) A vertical profile with less statistical significance can be estimated from averaging over a certain time period or equivalent height range ("slant profiles") on a slow, continuous ascent or descent. The definition of the time period is a trade-off between vertical resolution and statistical robustness. For basic turbulence parameters as wind speed variance, an averaging time of 60 s seems to be suitable to preserve the vertical structure while producing reasonable results. On the slant profile, turbulent fluctuations result from subtracting a Savitzky-Golay filtered data time series.

### 4.1.1 Dissipation rate

Siebert et al. (2006) discussed different methods to estimate the local dissipation rate from airborne in situ measurements. Here, the second order structure function

$$S^2(t^*) \equiv \overline{[u(t-t^*) - u(t)]^2}_\tau = 2 \cdot \varepsilon_\tau^{2/3} \cdot \left(t^* \cdot \overline{U}_\tau\right)^{2/3} \qquad (6)$$

is applied to estimate the local dissipation rate $\varepsilon_\tau$ for non-overlapping time periods of $\tau$. Averaged parameters in Eq. (6) are indicated by an overline and index $\tau$, $u(t)$ is the horizontal wind speed at the time $t$, $t^*$ is a time lag and $U$ is the overall wind vector. An averaging time of $\tau = 10$ s is defined, which is shorter compared to the wind speed variance averaging time. This is due to the fact that the dissipation rate is a small-scale, local parameter, whereas variances are influenced by the largest




contributing eddies. For each time period $\tau$, the structure function on the left side of Eq. (6) is evaluated for time lags $t^*$ in an empirical time range between $1/f_s = 0.2$ s and $1$ s. Fitting this curve to the right side of the equation yields $\varepsilon_\tau$ for each time period. If the structure function does not show a slope of $\varepsilon^{2/3}$ (fitted exponents in the range between 0.3 and 0.9 are accepted), as supposed for an inertial sub-range behavior (e.g., Wyngaard, 2010), no dissipation rate can be derived. Hence, the turbulent

structures are too small to be resolved with the applied sensors.

For the HP, local dissipation rates are calculated by applying the second order structure function to the measured horizontal wind vector $u(t)$. For UP measurements, the dissipation rate is calculated with the vertical wind component $w(t)$ and multiplied by $4/3$ for assuming isotropic turbulence.

### 4.1.2    Turbulent energy fluxes

To estimate turbulent fluxes, the eddy covariance method is used, by averaging covariances over a defined time period. Covariance of vertical wind speed $w$ and a second parameter leads to the equations for e.g. the turbulent heat flux and the momentum flux, which are commonly applied in airborne flux estimations (e.g., Wendisch and Brenguier, 2013). Here, we use the buoyancy flux $H_\mathrm{B}$ resulting from covariance of vertical wind speed $w$ and potential virtual temperature $\theta_v$, as the virtual temperature is directly measured by the sonic anemometer:

$$H_\mathrm{B} = \rho \cdot c_\mathrm{p} \cdot \overline{w' \cdot \theta_v'}, \tag{7}$$

where $\rho$ is the air density and $c_\mathrm{p}$ the specific heat capacity. Fluctuating parameters are marked with a prime. Turbulent energy fluxes can only be estimated from measurements with the UP, as the HP measurements do not provide the vertical wind vector.

The averaging time for applying Eq. 7 is restricted by external factors, but at best is long enough to capture the largest eddies contributing to the covariance. When averaging over the constant altitude segments of typically 10–15 minutes, it is not

possible to estimate statistically robust time averages of the turbulent fluxes. According to arguments given by Lenschow et al. (1994) and the conditions for our observation period (e.g., a low wind case with a mean horizontal wind speed of $2\,\mathrm{m\,s^{-1}}$ and integral time scales in the order of $\tau_w = 25$ s), averaging over around 200 minutes would be necessary to keep the statistical random error below 10 %. Such long averaging times would result in very few results due to the limited total flight time. Therefore, the constant altitude segments are restricted to relatively short time records of $T_\mathrm{m} = 620$ s with the advantage of

better resolving the vertical structure at a given time, instead of providing ensemble averages. The statistical flux error due to uncorrelated noise is $2\,\mathrm{W\,m^{-2}}$.

Besides averaging over time periods with constant altitude, the slant profile approach can be used by separating a continuous vertical profile into segments with a shorter averaging time. Here, we use $T_\mathrm{m} = 50$ s for the slant profiles. Using this approach, the magnitude of calculated fluxes cannot be compared to fluxes with a longer averaging time, but they provide an idea about

the vertical structure.





## 4.2 Derivation of radiative heating profiles

The local radiative heating rate is defined as the temporal change of temperature due to changes in the net irradiance $\partial F_{net}$ with the altitude $\partial z$ (Wendisch and Yang, 2012):

$$\frac{\partial T}{\partial t} = \frac{1}{\rho \cdot c_p} \frac{\partial F_{net}}{\partial z}, \tag{8}$$

where $\rho$ is the density of air and $c_p$ is the specific heat capacity at constant pressure. The net irradiance $F_{net}$ is defined as the difference of downward and upward irradiance:

$$F_{net} = F^{\downarrow} - F^{\uparrow}. \tag{9}$$

Combing Eq. 8 and Eq. 9 and substituting the derivative by the Taylor expansion results in the discrete form of the local heating rate $\zeta$ for a horizontal layer defined by top $z_t$ and bottom $z_b$ altitude

$$\zeta = \frac{\Delta T}{\Delta t} = \frac{1}{\rho \cdot c_p} \frac{\left(F_t^{\downarrow} - F_t^{\uparrow}\right) - \left(F_b^{\downarrow} - F_b^{\uparrow}\right)}{z_t - z_b}. \tag{10}$$

The thickness of this horizontal layer is defined as $\Delta z = z_t - z_b$. The magnitude of $\zeta$ strongly depends on the layer thickness considered for the calculation. In this respect, configuration 3 of the balloon operation (Fig. 2) provides two approaches to measure the net irradiance in the layer $\Delta z$: The collocated approach uses two identical BPs to calculate the heating rate for the layer between both platforms. The single-platform approach determines heating rates from analyzing the vertical profile of $F_{net}$ measured by a single BP.

For the collocated approach, $\Delta z$ changes only slightly with the drift of the balloon due to the tilt of the tether. As both platforms measure continuously in both altitudes, for each time step a radiative heating rate can be calculated. Thus, this approach is most suited to measure the time evolution of the heating rate in a layer with constant altitude. Profiles of $\zeta$ can be derived from ascents and descents of the balloon. The choice of the distance between both sensors considerably influences the calculated heating rate. Large $\Delta z$ average the heating rate for a thicker cloud layer and hence do not resolve small scale strong extrema within the layer. Short distances between both platforms result in small differences in $F_{net}$, which may result in high uncertainties in the estimated heating rates. In addition, both instruments can interact mechanically and influence each other's measurements, when placed too close to each other. Therefore, a minimum distance of $10\,\mathrm{m}$ is required. Further, the cloud top and thus the cloud top cooling varies in a range of $\pm 30\,\mathrm{m}$. Therefore, both platforms are separated by at least $60\,\mathrm{m}$ during the PASCAL observation period.

The single-platform approach can provide a higher vertical resolution of the heating rate. The layer $\Delta z$ is given by measurements of consecutive times steps. However, for the single-platform approach, the environment is assumed to be in steady state when profiling the vertical column. As the boundary layer and especially the cloud top may vary in a short time, the $F_{net}$ profile is post processed to account for these cloud inhomogeneities. The profile is separated into $5\,\mathrm{m}$ layers and the maximum of each layer is chosen to represent cloudy conditions. The resulting profile is additionally smoothed with a window length of $15\,\mathrm{m}$.





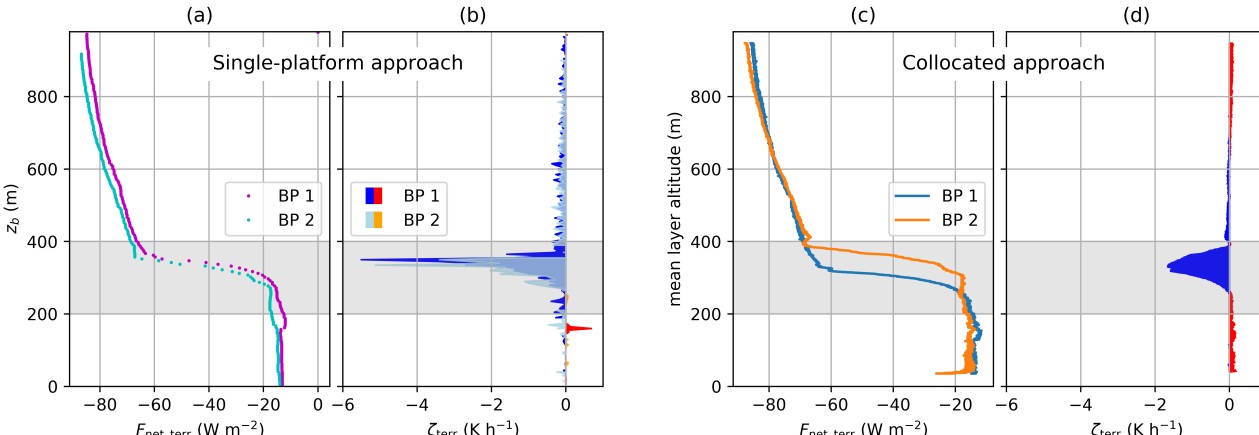

**Figure 9.** Profiles of net terrestrial irradiance measured on 5 June 2017, 17:00 UTC with respect to the individual sensor altitude (a) and with respect to the mean altitude of the layer between both platforms (c). The calculated heating rates are shown for both individual sensors using the single-platform approach (b) and combining both sensors in the collocated approach (d). Positive values of the heating rate are in red/orange color (indicating a warming), negative values in blue/light blue (indicating a cooling).

Figure 9 shows an example for the net terrestrial irradiance measured by both BPs and the derived radiative heating rates during the second flight on 5 June 2017 for the single-platform approach (a, b) and the collocated approach (c, d). For this flight, both sensors are separated by 65 m. Panel a displays the terrestrial heating rate $F_{\mathrm{net,terr}}$ with respect to the individual altitude of each sensor referring to the single-platform approach, while panel c shows $F_{\mathrm{net,terr}}$ with respect to the mean altitude

of the layer between both sensors, which refers to the collocated approach. All profiles show a decrease in $F_{\mathrm{net,terr}}$ at cloud top by about $50\,\mathrm{W\,m^{-2}}$. The individual profiles (panel a) are slightly departed in altitude by about $10\,\mathrm{m}$. This indicates a slight decrease of the cloud top height during the time between the ascent of BP 1 and BP 2 through the cloud top layer. With an ascent rate of $1\,\mathrm{m\,s^{-1}}$ the penetration of the sensors through the same layer is separated in time by about $60\,\mathrm{s}$ during which the cloud obviously changed. This difference in the position of the cloud top is transferred into the calculated heating rate profiles

shown in Fig. 9b. The general pattern of the profiles agrees for both platforms, with the minimum radiative heating rate being situated below the cloud top. A maximum cooling of $5.5\,\mathrm{K\,h^{-1}}$ is observed for the single platform approach. Figure 9d shows the heating rate profile using the collocated approach, which is less affected by the cloud inhomogeneity due to the larger thickness of the corresponding vertical layer. For the thicker layer analyzed by the collocated approach (compare Eq. 10), the maximum radiative cooling is reduced to $1.5\,\mathrm{K\,h^{-1}}$, but spreads over a wider altitude range. Below cloud base, the collocated

approach shows a slight warming of the sub-cloud layer by $0.2\,\mathrm{K\,h^{-1}}$, which is not visible from the single-platform approach profiles.





## 5  Measurement examples from PASCAL

During PASCAL, the tethered balloon was operated from an ice floe north of Svalbard at around 81.8° N in the period
5 to 14 June 2017. On the ice floe, the balloon site was established 200 m from the ship, close to a 10 m high mast for con-
tinuous meteorology and turbulence measurements. The tethered balloon was launched 16 times within nine operating days. A
list of all launches showing the individual configuration and the general meteorological conditions is given by Wendisch et al.
(2018).

The observation period on the ice floe was characterized by a warm maritime air mass (Knudsen et al., 2018). Cloud condi-
tions were highly variable with an average cloud fraction of 65 %. Here, three case studies for observations in different ABL
and cloud conditions are presented. The first case observed on 5 June 2017 comprises a single, low cloud layer, the second case
on 10 June 2017 represents a cloudless day, whereas the third case observed on 14 June 2017 is characterized by a lower cloud
layer with multiple cloud layers aloft. This section aims to demonstrate the potential of the new BELUGA set-up. Scientific
questions building on those measurements will be elaborated in upcoming publications.

### 5.1  A single layer cloud: 5 June 2017

The ABL stratification observed on 5 June is characterized by a single, 200–300 m thick stratocumulus cloud layer close to the
ground. This cloud type was typical for most days of the measuring period. For the presented case, instrument configuration 1
(Fig. 2) with the UP and one BP was applied. The time series of the balloon altitude is shown in Fig. 3; it includes a continuous
profile followed by constant altitude segments. The cloud base and top height varied during the observation period due to
spatial and temporal cloud heterogeneities. Therefore, it is challenging to exactly determine cloud top and base altitude by the
profiles of the balloon observations only. A combination of Cloudnet data (lidar and radar data measured on the research vessel;
Griesche et al., 2019), the relative humidity and location of the maximum cloud top cooling from the balloon observations is
used to define the cloud extend for the vertical profiles.

The vertical structure of the ABL and the derived turbulent and radiative parameters are shown in Fig. 10. The vertical
profiles are derived from the first continuous ascent from ground up to 1 km altitude. The ABL is characterized by a strong
potential temperature inversion of ∼10.5 K within a 120 m thick vertical layer, capping the cloud layer and a nearly neutral
sub-cloud layer. Above the inversion, the mean potential temperature slightly decreases with height, but shows several layers
with potential temperatures varying between 8°C and 14°C. Relative humidity is around 90 % below the cloud and decreases
to 40 % above the cloud layer.

The terrestrial heating rate calculated by the single-platform approach is shown in Fig. 10b. A maximum cloud top cooling
of 5.5 K h$^{-1}$ is observed within the uppermost 2 m of the cloud. At cloud base, a weak warming of the lowest 30 m of the
cloud layer by 1 K h$^{-1}$ is observed. This cloud base heating indicates that at cloud base the terrestrial radiation emitted by the
warmer underlying surface exceeds the emission by the colder cloud base.

Horizontal wind speed is generally low with values around 2 m s$^{-1}$ inside the cloud and 1 m s$^{-1}$ above. A wind speed
maximum is observed in the upper half of the cloud, where the vertical wind speed is positive (directed upward), whereas





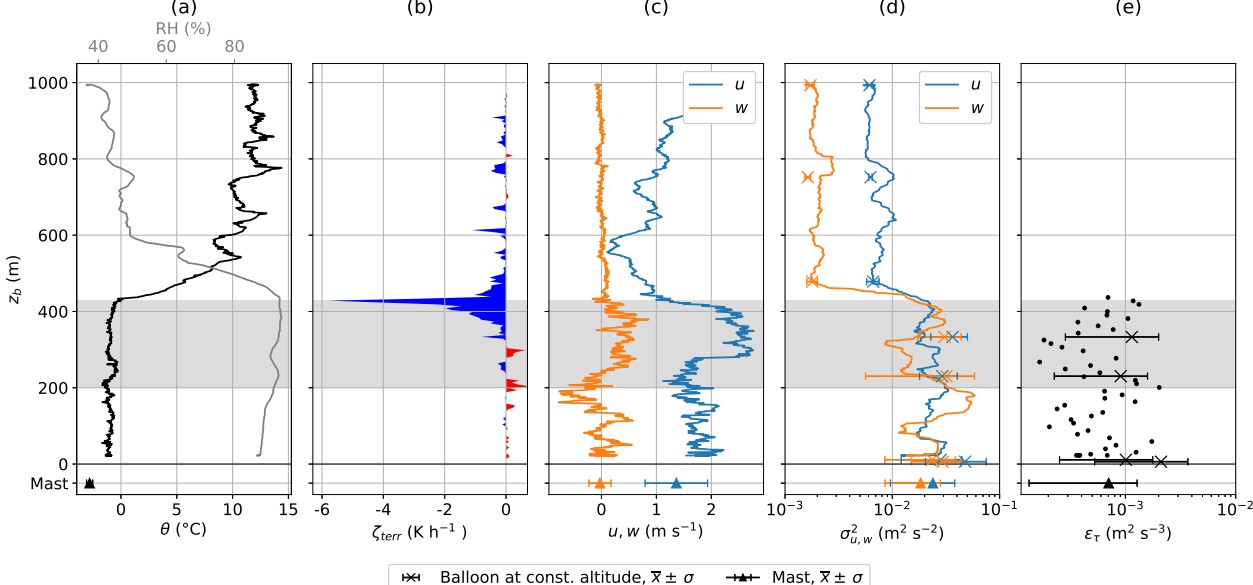

**Figure 10.** Vertical profiles of potential temperature $\theta$ and relative humidity RH (a), terrestrial heating rate $\zeta_{\mathrm{terr}}$ (b), horizontal wind $u$ and vertical wind $w$ (c), local wind speed variance $\sigma_{u,w}$ (d) and the local dissipation rate $\varepsilon_\tau$ (e) observed on 5 June 2017. Solid lines and small dots show slant profile data measured during the ascent (60 s running time window for variances, non-overlapping time segments of 10 s for dissipation rate). Crosses and corresponding horizontal bars in panel d and e show mean and standard deviation of dissipation rate and variances for measurements at constant altitude (segments of typically 10 minutes) obtained during the descent. Averaged ground-based measurements at the meteorological mast are shown as triangles with the standard deviation indicated by horizontal bars. The cloud extend is indicated by the gray shaded area.

it turns negative at the cloud base. Figure 10d shows the vertical profile of variances for horizontal and vertical wind speed obtained from the continuous ascent (solid lines) and measurements at the constant altitude segments (crosses). Variances are averaged over 60 s for both methods (consecutive intervals on the 10 minutes segments), resulting in a similar vertical structure and comparable magnitude. Maximum variances are observed below the inversion layer, with local maxima near cloud top, cloud base and at the ground. The difference between in-cloud and above-cloud variance for vertical wind speed is one order of magnitude. Variances for vertical and horizontal wind speed show similar values below the inversion. However, within and above the inversion layer vertical fluctuations are rather diminished as compared to the horizontal wind fluctuations.

Figure 10e presents local dissipation rates $\varepsilon_\tau$ calculated for the slant profile, the constant altitude segments and the mast data. $\varepsilon_\tau$ is in the order of $10^{-3}$ $\mathrm{m^2\,s^{-3}}$ with high variability below the inversion. Above the inversion, the turbulence intensity is too low and $\varepsilon_\tau$ cannot be reasonably estimated. The majority of data derived from the slant profile are within the standard deviation of the statistically more reliable measurements within constant altitude segments. This indicates, that the measurements of slant profiles are applicable to estimate local dissipation rates.





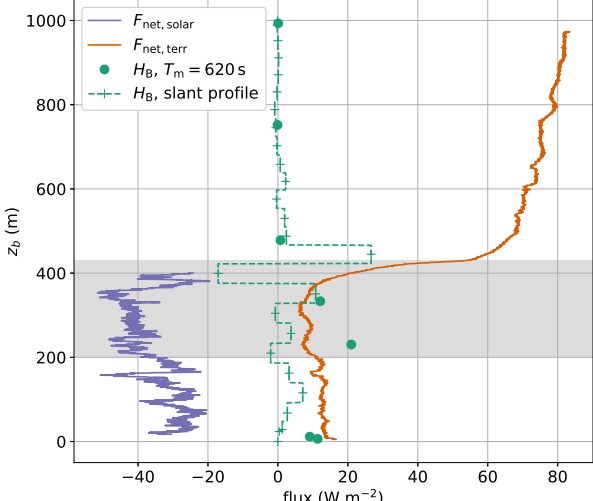

**Figure 11.** Vertical profile of buoyancy flux $H_B$ and solar and terrestrial net irradiance $F_{net}$ on 5 June 2017. Balloon-borne buoyancy fluxes are calculated from 620 s time periods at constant altitudes (filled circles) and the slant profile with $T_m = 50$ s (crosses with dashed line). Positive values indicate upward oriented fluxes.

In the present case, no clear indication for a decoupling between a cloud mixed layer and a surface mixed layer is given, which is in contrast to the observations by Brooks et al. (2017), who suggest that a decoupling is typical for the Arctic ABL in summer. The vertical profile of dissipation rate suggests a continuous mixed layer from ground to cloud top. In contrast, the change in horizontal wind speed and its variance allow to assume a separate sub-cloud mixed layer with wind shear driven

turbulence. However, at least around cloud top the increased turbulence can be explained by the cloud top cooling.

The combined turbulence and radiation measurements allow to compare irradiances and turbulent heat fluxes and to calculate the energy budget in different altitudes. For this purpose, the vertical profile of the net solar and terrestrial irradiances and the buoyancy flux observed on 5 June 2017 are shown in Fig. 11. The components of the radiation budget are significantly larger than the turbulent fluxes and dominate the energy balance. Contrarily to Eq. 9, here the net irradiance is defined as

difference between upward and downward irradiance which is required to have a consistent definition of all fluxes and allows a calculation of the energy budget. Therefore, the upward directed net terrestrial irradiance is defined as positive and is almost constant between ground and close to the cloud top. At cloud top, the net terrestrial irradiance increases from 10 W m$^{-2}$ to 50 W m$^{-2}$, and then slowly continues increasing with height. The solar irradiance is fairly constant below and within the cloud in a range between -20 W m$^{-2}$ and -50 W m$^{-2}$. The buoyancy flux obtained from the constant altitude segments is

most pronounced inside the cloud with a maximum near cloud base of 20 W m$^{-2}$. Above the cloud, the buoyancy flux is close to zero. Near ground, the buoyancy flux is around 10 W m$^{-2}$. A sequence of buoyancy flux estimates on the mast with the same averaging time results in a similar mean value of 13 W m$^{-2}$, but with high temporal variability ($\sigma_H = 15$ W m$^{-2}$).





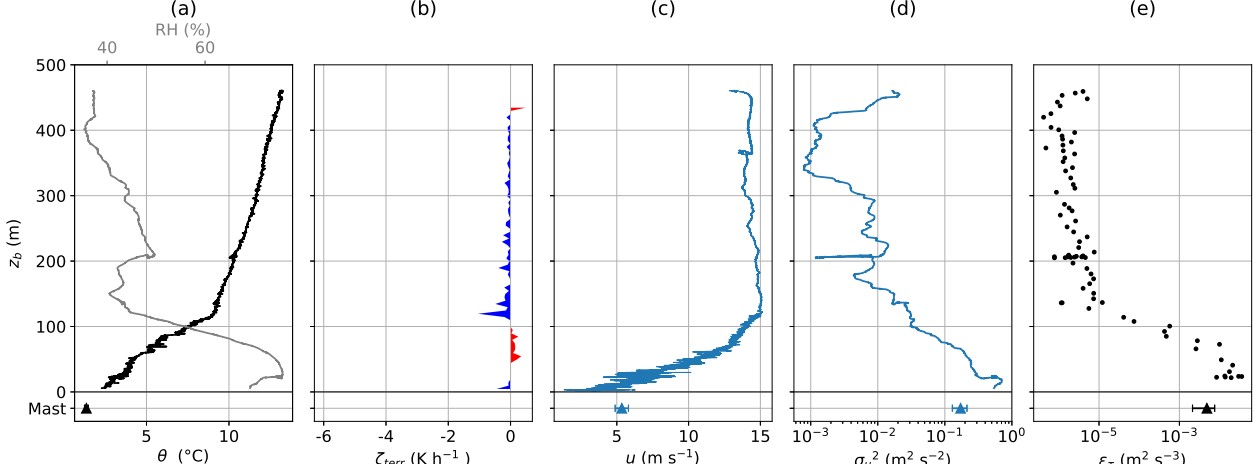

**Figure 12.** Same as Fig. 10, but for an ascent on 10 June 2017. The HP was applied to measure the horizontal wind vector $u$ and no constant altitude segments were recorded.

The magnitude of the buoyancy flux derived from the slant profile is significantly smaller than the fluxes measured at constant altitude, but a strong positive and negative maximum are observed at cloud top.

## 5.2 Cloudless case: 10 June 2017

For comparison, measurements collected during a cloudless day with strong wind speed are presented. Due to the strong wind,
only the HP and one BP (configuration 2) were deployed in order to reduce the payload. The balloon drifted horizontally around 500 m while not exceeding 500 m altitude. Figure 12 shows the vertical profile of measured and calculated parameters obtained during a continuous ascent. The ABL exhibits a constant surface-based potential temperature inversion of 6.5 K/ 100 m between ground and 120 m altitude. The layer above still exhibits a stable stratification up to the maximum altitude. Relative humidity is around 80 % near ground and decreases within the inversion to 40–50 %.

Terrestrial heating rates are close to zero throughout the whole vertical profile with a change in sign at the inversion layer top height. Above the inversion, a weak radiative cooling is observed with a maximum of $1\,\mathrm{K\,h^{-1}}$ at the top of the inversion layer. Within the inversion layer, a slight warming of $0.5\,\mathrm{K\,h^{-1}}$ is present.

The horizontal wind speed increases from $5\,\mathrm{m\,s^{-1}}$ near the ground to $15\,\mathrm{m\,s^{-1}}$ within the inversion layer and is nearly constant above. In contrast, the wind speed variance constantly decreases with altitude. Turbulent dissipation is about $0.1\,\mathrm{m^2\,s^{-3}}$
near ground and decreases inside the inversion to very small values. The layer above reveals very weak turbulence, despite the high wind speed, but without significant wind shear. This example demonstrates the capability of the hot-wire probe to resolve energy dissipation rates down to below $10^{-6}\,\mathrm{m^2\,s^{-3}}$.





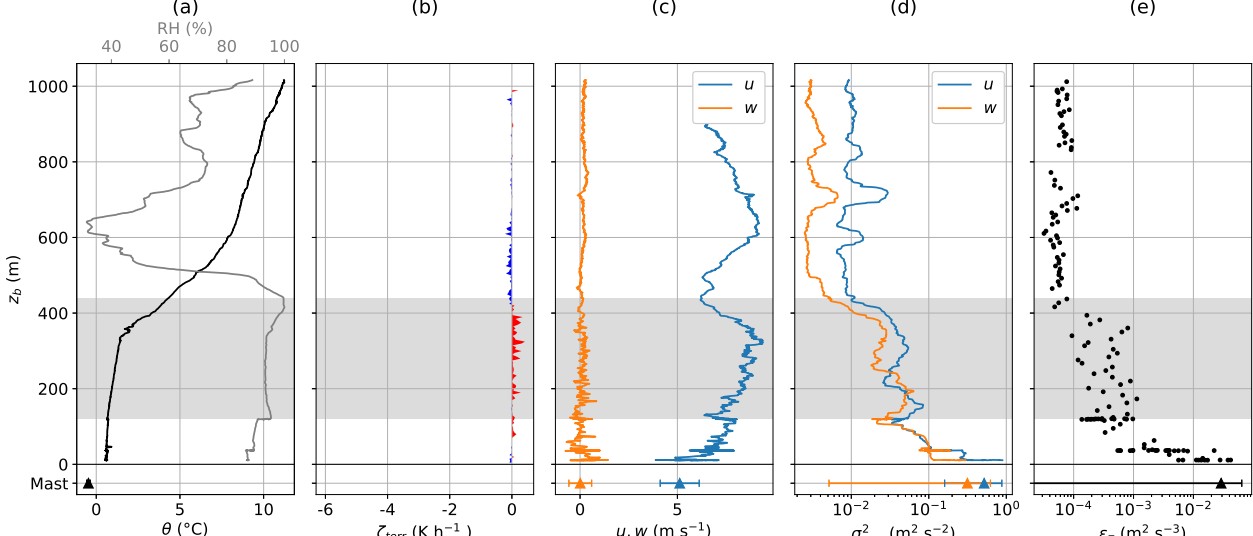

**Figure 13.** Same as Fig. 10, but for an ascent on 14 June 2017. No constant altitude segments were recorded.

In the presented case, the ABL exhibits a highly turbulent layer inside the ground-based temperature inversion, which is probably induced by wind shear. As measurements were taken with the HP, no turbulent flux estimation is available.

### 5.3 Multi layer clouds: 14 June 2017

On 14 June, a low stratocumulus cloud layer up to a height of 500 m was observed with multiple cloud layers extending from
1.2 km to 4 km altitude, and topped by a cirrus cloud. The UP and two BP (configuration 3) were applied to measure two continuous vertical profiles up to 1000 m between 9:00 and 11:00 UTC.

The boundary layer structure and derived turbulent and radiation parameters are illustrated in Fig. 13. The lower stratocumulus cloud is topped by a 7 K, 250 m thick inversion layer. However, the relative humidity profile suggests that the cloud penetrates into the inversion layer. Terrestrial heating rates are fluctuating close to zero throughout the whole profile, with a
sightly positive tendency below cloud top and a negative tendency above. This indicates a slight warming of the layer below cloud top, similar to the warming of the inversion layer in the cloudless case.

Horizontal wind speed increases from $6 \, \mathrm{m \, s^{-1}}$ near the ground to a maximum of $8 \, \mathrm{m \, s^{-1}}$ shortly below cloud top. A second maximum is observed in 600 m just above the inversion layer. In contrast to wind speed, its variance decreases from ground to cloud top. The same applies for the dissipation rate with maximum values of almost $10^{-1} \, \mathrm{m^2 \, s^{-3}}$ in the lowermost 30 m,
relatively constant values inside the cloud and less than $10^{-4} \, \mathrm{m^2 \, s^{-3}}$ above the cloud. In the region of the upper wind speed maximum, variances and dissipation rate show a small increase probably due to wind-shear induced turbulence. In this case, turbulence is not induced by cloud top cooling. Instead, increased turbulence at ground level is probably the consequence of wind shear near the surface.





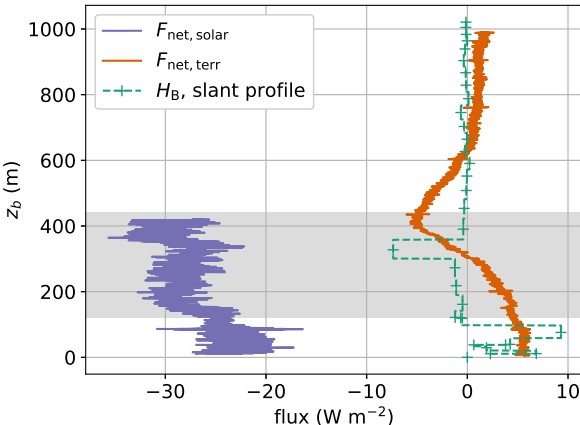

**Figure 14.** Solar and terrestrial net irradiance $F_{net}$ and buoyancy flux $H_B$ on 14 June. The buoyancy fluxes are based on the slant profile. The shaded area represents the lower cloud layer.

Figure 14 illustrates the vertical profile of turbulent fluxes and net irradiances. Solar and terrestrial irradiances are of smaller magnitude, and less influenced by the low cloud layer than on 5 June 2017. The terrestrial irradiance shows an upward directed flux of $5\,\mathrm{W\,m^{-2}}$ near the surface, and decreases towards cloud top to negative values. Above the cloud, the terrestrial flux is close to zero. Due to the cloud layer above, the terrestrial radiation emitted by the top of the lower cloud layer and by the base

of the upper cloud layer are almost balanced. The buoyancy flux fluctuates around zero throughout the whole profile with a positive peak below the cloud layer and a negative peak inside the cloud. To conclude, in the presence of higher cloud layers, the lower cloud layer has a significantly smaller influence on radiative and turbulent parameters.

## 6   Summary and discussion

Measurements with research aircraft in supercooled Arctic clouds are challenging, mainly due to icing induced risks. Tethered

balloons are less prone to icing mainly due to two facts: (i) they move at a lower true airspeed resulting in less accumulation of ice, and (ii) the probability of ice sticking on the balloon is reduced due to its flexible envelope. Another main advantage of tethered balloons is that they provide almost truly vertical profiles of measurements. These vertical profiles enable to study small scale local cloud and atmospheric properties, which are otherwise smoothed out by the large distances covered by aircraft. In addition, tethered balloon systems can observe individual profiles within a single turbulent eddy, whereas aircraft

measurements average over at least a few eddies. However, the difference in true airspeed of balloon and aircraft measurements results in variable statistics: an aircraft can probe a much larger area enabling more robust statistics. Also, the aircraft is more flexible in space. Nevertheless, for many research questions it is crucial to obtain vertical profile measurements starting at ground level. This is particularly true for the Arctic ABL, which is often shallow and characterized by height dependent energy fluxes in the lowermost part. To understand complex atmospheric processes interacting at the sea ice interface, it is crucial





to measure the lowermost part of the ABL, which can be realized by applying a tethered balloon. For the reasons mentioned above, the tethered balloon system BELUGA was developed for measurements in supercooled clouds.

BELUGA is a modular tethered balloon system and comprises of the tethered balloon itself and multiple instrument packages for high resolution and collocated in situ vertical profiling. The flexible combination of the instruments allows to pursue specific

scientific goals and to adapt to different environmental conditions. The instruments are carried by a $90\,\mathrm{m}^3$ helium-filled tethered balloon, which has proven to reliably operate in the Arctic environment including cloudy conditions, wind speeds of up to $15\,\mathrm{m\,s^{-1}}$ and light icing. During the first application of BELUGA within the PASCAL campaign (Wendisch et al., 2018; Knudsen et al., 2018), the operation of the balloon was not seriously affected by icing. A small amount of riming could be removed mechanically by hand and by the deflection pulleys ahead of the winch. In situations with more icing, the payload

weight was reduced accordingly, resulting in more free lift compensating the additional accumulated weight of ice or snow on the balloon envelope. The same holds true for stronger wind conditions, where more free lift of the balloon resulted in more stable flight conditions. Typically, $30\,\%$ of the free lift at ground (about $8\,\mathrm{kg}$ with a free lift of $25\,\mathrm{daN}$ for the balloon) were used for the instrumental payload under strong wind conditions, whereas in almost calm wind conditions the payload could be increased up to 10 to $12\,\mathrm{kg}$. The typical ceiling of the balloon is about $1.5\,\mathrm{km}$.

Currently, three instrument packages are available for the study of turbulent and radiative parameters including energy fluxes: (i) an ultrasonic based turbulence probe, which measures the three-dimensional wind vector for turbulence observations including vertical turbulent energy fluxes, (ii) a small and lightweight hot-wire based turbulence probe, which allows for energy dissipation rate measurements, and (iii) an upward and downward looking broadband radiation package, which allows net irradiance measurements and the determination of radiative heating rates. Collocated measurements of turbulent and radiative

fluxes can be combined to link cloud top radiative cooling, which is associated with negative buoyancy fluxes, and turbulent mixing.

After a technical introduction of the three instrument packages including a description of their performance and limitations, the methods to calculate and analyze turbulent and radiative properties in cloudy conditions are introduced. The capability of the new system is illustrated by three measurement examples observed during the two-week PASCAL sea-ice drift period.

During PASCAL, the tethered balloon was operated from an ice floe under a variety of different meteorological conditions. The presented examples describe (i) a single cloud layer, (ii) a cloudless situation, and (iii) a multi layer cloud case. The BELUGA measurements during these three atmospheric situations emphasize the value of collocated measurements with a high vertical resolution.

This work aims to demonstrate the potential of the new tethered balloon set-up. Scientific questions building on those mea-

surements will be elaborated in upcoming publications. Further instrument packages are under development, including a comprehensive aerosol and cloud microphysical sensor system. Based on the observations during PASCAL and future deployment of BELUGA, the following scientific questions will be pursued:

- How are turbulence/ radiation vertical profiles influenced by typical ABL structures, different cloud properties and aerosol loads?




- How do vertical profiles influence surface radiative forcing and cooling/ warming?

- How are processes at cloud top influenced by turbulent mixing, heating rates and humidity sources?

- How does the ABL structure vary under cloudless conditions?

- To what extend do cloud microphysical properties influence ABL properties?

5 *Data availability.* Data related to the present article are available in Open Access through PANGAEA – Data Publisher for Earth & Environmental Science: Balloon-borne data at https://doi.org/10.1594/PANGAEA.xxxxxx and mast data at https://doi.org/10.1594/PANGAEA.xxxxxx.

*Author contributions.* UE and MG developed the instruments, performed the measurements, analyzed the data and drafted the manuscript. HS was responsible for the overall balloon system. HS, MW and AE contributed to the data analysis and the writing of the manuscript.

*Competing interests.* The authors declare that they have no conflict of interest.

10 *Acknowledgements.* We gratefully acknowledge the funding by the Deutsche Forschungsgemeinschaft (DFG, German Research Foundation) – Project Number 268020496 – TRR 172, within the Transregional Collaborative Research Center "ArctiC Amplification: Climate Relevant Atmospheric and SurfaCe Processes, and Feedback Mechanisms (AC)³" in sub-project A02. We highly appreciate the participation in RV *Polarstern* cruise PS 106.1 (expedition grant number AWI-PS106-00) and want to thank the crew and supporting scientists on the cruise, especially Thomas Conrath, Felix Lauermann and Kai-Erik Szodry.





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
