# Peer review of "The new BELUGA setup for collocated turbulence and radiation measurements using a tethered balloon: First applications in the cloudy Arctic boundary layer"

_Atmospheric Measurement Techniques, 2019_

## Referee Comment (RC1) · Ian Brooks (Referee) · 9 Apr 2019

Review of: AMT-2019-80
**"The new BELUGA setup for collocated turbulence and radiation measurements using a tethered balloon: First applications in the cloudy Arctic boundary layer"**
**Ulrike Egerer, Matthias Gottschalk, Holger Siebert, André Ehrlich, and Manfred Wendisch**

This paper presents an overview of a new balloon-borne measurement system for atmospheric boundary layer research with several, modular, component instrument packages. The measurement systems are well documented, and their capabilities nicely demonstrated through presentation of measurements from 3 different Arctic boundary layer cases.

The paper is clear and well written, and should be acceptable for publication with only minor revisions – mostly for clarity or corrections of minor typographical issues. These are documented below.

**Detailed comments:**

Line 11: 'The majority of Arctic clouds is located...' -> 'The majority of Arctic clouds are located...'

Line 14: The phrasing 'In most climate models, turbulent and radiative fluxes in these low altitudes are underrepresented,' is ambiguous. This could mean 'under estimated', ie too small. Or 'poorly represented' - I think the intended meaning is the latter, and should be rephrased to make this clear.

Line 26: '...vertical resolution of typically 45m...' – 'typically' is perhaps the wrong work, imply most frequent. The resolution is instrument dependent, so varies, but is usually a few 10s to ~100m.

Line 29: 'Tethered balloon measurements enable to bridge the gap between surface based and aircraft measurements...' -> 'Tethered balloon measurements enable the gap between surface based and aircraft measurements to be bridged...'

With respect to the discussion of UAV and tethered balloon systems in general, the recent paper by de Boer et al. 2018, (doi: 10.1175/BAMS-D-17-0156.1) is a useful reference, covering both tethered balloon and UAV measurements in Arctic Alaska.

Line 11: '...included substantial instrumentation of different research groups...' -> '...included substantial instrumentation from different research groups...'

Line 16: '...new setup are...' -> '...new system are...'

Line 18: '...discussion of limitation and...' -> '...discussion of the limitations and...'

Line 23: '...between the ground and...' -> '...between the surface and...' (pedantic, but your example data is over a sea ice surface, not 'the ground')

Line 26: '...and at light icing...' – 'at' should be 'in' or 'under'

Line 29: 'The packages can be deployed on the balloon considering three main configurations of turbulence and radiation measurements' – slightly awkward phrasing, suggest ' The packages are deployed in one of three main configurations, depending on the conditions and requirements for turbulence and radiation measurements'

Line 5: '...to adjust...' -> '...adjustment of...'

Line 9: The quoted 'standard lapse rate' of 6.5 k km$^{-1}$ is rather low, the dry lapse rate is approximately 9.8 K/km (AMS glossary), and at freezing point in just saturated air is 9.968 K/km. The value given here is approximately the wet adiabatic lapse rate. Ideally the wet/dry value should be used in/out of cloud to give most accurate height.

Line 18: 'motion of the balloon within the' -> 'motion of the balloon with the'

Figure 8 caption: 'measured on the mast in a height of...' -> 'measured on the mast at a height of...'

Line 12: move the value for uncorrelated noise for the balloon out of equation line and into text – as it is, it's easy to miss the value in equation, and text then appears to read that it can be calculated for the balloon, but only quoted for mast.

Line 1: '...a slight flatten or...' -> '...a slight flattening or...'

Line 10: 'Finally, the same filter algorithm is...' -> 'Finally, the same filter algorithm was...'

Line 24: regarding the averaging time for fixed altitude flux estimates. While the 620s periods is not unreasonable, it may be too short under some conditions. It is worth examining cospectra and ogives curves to evaluate whether all the low-frequency flux contributions are captured, and if not estimate the missing flux contribution. 30 minutes would be long enough to capture all the flux contributions under most conditions, but clearly must be traded off against the available time and number of levels you can sample.

Line 21: '...cloud extend for...' -> '...cloud extent for...'

Line 33: in discussion of vertical wind, it is worth noting here that these regions of significant upward/downward motion cannot be mean motions, but the sampling of up/down moving portions of a large, boundary-layer scale eddy. It is notable, however, that their vertical location coincides with the regions of higher/lower horizontal wind. Possibly this too is sampling of a large scale eddy?

A minor issue of phrasing here (and in discussion of standard deviation of w) – the phrase 'wind speed' tends to imply a mean quantity, and thus 'vertical wind speed' seems inappropriate for w, for which simply 'vertical velocity' is perhaps better terminology.

Figure 10 caption, last line: 'the cloud extend...' -> 'the cloud extent...'

Figure 11 caption. Here and elsewhere I think the term 'terrestrial' to refer to the infra red irradiance is misleading, since much of the infra red budget has nothing to do with the surface. 'infra red' would be preferable.

Add an explicit reference to '$H_B$' in the caption for buoyancy flux.

Line 10: the phrasing of "Contrarily to Eq. 9, here the net irradiance is defined as difference between upward and downward irradiance which is required to have a consistent definition of all fluxes and allows a calculation of the energy budget. Therefore, the upward directed net terrestrial irradiance is defined as positive" is awkward. Suggest rephrasing as: "In contrast to Eq. 9, here the net irradiance is defined at the difference between the upward and downward components, and in order to maintain consistency, we define upward fluxes as positive for all fluxes, radiative and turbulent."

Figure 11 and 14 – why do the solar irradiances stop at cloud top, while the infra red values extend above?

Line 2 & figure 11. The strong negative buoyancy flux just below cloud top might be associated with entrainment. It is hard, however, to find a physical explanation for a strong upward flux just above cloud top – this might be an artefact of calculating an eddy covariance flux from a slant profile that crosses a strong and changing gradient at the inversion where perturbations from the 'mean' can result from changes in the mean rather than true turbulent fluctuations. I would treat this value with extreme caution.

Line 6 – the negative peak in buoyancy flux 'inside the cloud' is located just below the inversion base – again possibly associated with entrainment mixing, though again, calculating a flux from a slant profile across a strong and changing gradient is problematic, and I would want to carefully select the portion of the profile used to stay below the inversion base.

---

## Referee Comment (RC2) · Anonymous Referee #2 · 31 May 2019

This paper documents the setup and capabilities of the new tethered balloon "BELUGA" and presents results from flights that measured a single layer cloud, a multi-layer cloud, and clear-sky conditions during the PASCAL campaign. The setup currently includes instruments that measure winds, temperature, relative humidity, turbulent fluxes, and broadband radiative fluxes. The balloon can fly up to 1.5 km. Instruments are currently being developed to measure aerosols and cloud microphysics.

This paper is carefully written and provides a detailed description of the new modular tethered balloon system. This is a very important development that will provide new

insights into Arctic boundary layers and will provide the measurements needed to validate and improve turbulent and microphysical model parameterizations. I believe this paper is suitable for publication after minor revisions.

Comments:

1) Page 1, line 5-6: "Collocated data acquisition allows for estimates of the driving parameters in the energy balance at various heights."

2) Page 2, line 11: "The majority of Arctic clouds are located with the ABL."

3) Page 5, Figure 2 caption: "HW" should read "HP".

4) Page 6, line 11: "The data streams are synchronized by an analog. . ."

5) Page 18, line 6: ". . .are slightly separated in altitude..."

6) Page 19, line 2: ". . .north of Svalbard around 81.8N during 5-14 June 2017."

7) Page 20, Figure 10 caption: "The cloud extent is indicated by gray shading."

8) Page 21, line 3-5: Given the uncertainty in the estimates it is not clear that a near-surface mixed layer can be distinguished from the main cloud-driven mixed layer.

9) Page 21, line 9: "Contrary to Eq. 9. . ."

10) Page 24, line 12: ". . .they provide near-vertical profiles of collocated measurements. These vertical profiles enable the study of . . ."

11) Page 25, line 4: ". . .instruments allows the pursuit of specific. . ."

---

## Author Comment (AC1) · 21 Jun 2019

**Author's response to Ian Brooks' review of: AMT-2019-80**

We thank Ian Brooks for his constructive review, which significantly improved the quality of the manuscript. The author's response is structured as follows:

**Comments of reviewer**
Author's answer to the comment
*"New text passage"*

General remarks

- All suggestions for rephrasing and including references were implemented. This remarkably improved the language quality and made the manuscript more comprehensive.
- The term buoyancy flux, which was used throughout the discussion paper, is misleading. For the flux calculation, we used the definition of the sensible heat flux, but with the potential virtual temperature instead of the potential temperature. We renamed „*buoyancy flux $H_B$*" as „*virtual heat flux H*". This term was introduced by Angevine, 1993 (doi: 10.1175/1520-0450(1993)032<1901:VHFMFA>2.0.CO;2) and the reference is now included in the manuscript.

Answers to specific reviewer comments

- **Page 4, Line 9: The quoted 'standard lapse rate' of 6.5 k km-1 is rather low, the dry lapse rate is approximately 9.8 K/km (AMS glossary), and at freezing point in just saturated air is 9.968 K/km. The value given here is approximately the wet adiabatic lapse rate. Ideally the wet/dry value should be used in/out of cloud to give most accurate height.**
  You are right, but here we use the value given for the ICAO standard atmosphere (z < 11 km), as referred to in Wendisch & Brenguier 2013 (doi: 10.1002/9783527653218, p. 11). With this, we want to standardize the barometric height calculation, independently of the presence of clouds, and make it comparable to the aircraft heights. However, we compared also to GPS height and the differences were in the range of ±10 m below 1.000 m altitude.

- **Page 16, Line 24: regarding the averaging time for fixed altitude flux estimates. While the 620s periods is not unreasonable, it may be too short under some conditions. It is worth examining cospectra and ogives curves to evaluate whether all the low-frequency flux contributions are captured, and if not estimate the missing flux contribution. 30 minutes would be long enough to capture all the flux contributions under most conditions, but clearly must be traded off against the available time and number of levels you can sample.**
  That's a good point. We calculated the co-spectrum and ogive for a 1h period (5 June 2017) with the balloon kept near the ground, associated with a high degree of turbulence. The ogive suggests, that after 620s, 91% of the flux value are captured. Further, the flux averaged over the whole period is close to the mean value calculated from 44 (overlapping) 620s-segments within the whole period. From this, we conclude, that the 620s period results in an error in the order of 10%. For higher altitudes, we expect the integral time and length scales to decrease which in turn - assuming a similar correlation coefficient between vertical velocity and temperature - results in a decreasing statistical error. We also estimated the statistical error based on Lenschow's arguments already in the manuscript and now include the resulting statistical error for the 620s period.

  *"Applying this averaging time, the random flux error is around 40% in the turbulent regions."*

[Figure]

- **Page 19, Line 33: in discussion of vertical wind, it is worth noting here that these regions of significant upward/downward motion cannot be mean motions, but the sampling of up/down moving portions of a large, boundary-layer scale eddy. It is notable, however, that their vertical location coincides with the regions of higher/lower horizontal wind. Possibly this too is sampling of a large scale eddy?**
  A first idea was that this might result from an artefact in the wind vector transformation. For the balloon, the wind/ turbulence and the motion of the instrument itself are closely connected, which makes the transformation challenging. Therefore, we verified again the transformation of the wind vector during the ascent and could not find, that the vertical wind variation results from a measurement artefact (e.g., an offset in the pitch angle). We are now confident that the shown wind velocity components are correct. Your interpretation of sampling within one large-scale eddy is convincing and in the manuscript we made a short comment about this possible explanation:
  *"As the location of upward motion and increased horizontal wind coincides, this might be an indication for a large, boundary-layer scale eddy."*

  **A minor issue of phrasing here (and in discussion of standard deviation of w) – the phrase 'wind speed' tends to imply a mean quantity, and thus 'vertical wind speed' seems inappropriate for w, for which simply 'vertical velocity' is perhaps better terminology.**
  "*Wind speed*" was replaced by "*wind velocity*", when it did not refer to the mean value.

- **Page 21, Figure 11 caption. Here and elsewhere I think the term 'terrestrial' to refer to the infra red irradiance is misleading, since much of the infra red budget has nothing to do with the surface. 'infra red' would be preferable.**
  As our analysis focuses mostly on clouds, the term 'terrestrial' might truly be misleading. Thus, we will change the term to *"thermal infrared" (TIR)*. According to the definition of TIR from ~3-50 µm (Manfred Wendisch and Ping Yang, 2012), this range will closely agree to our measured values of 4.5-42 µm.

- **Figure 11 and 14 – why do the solar irradiances stop at cloud top, while the infra red values extend above?**
  The misalignment of the solar sensors exposed to direct sunlight are usually corrected using the method described by Bannehr and Schwiesow (1993). In case of a strong

movement and misalignment of the sensor package, the correction could not be applied successfully and a bias remained. These conditions with strong wind shear where predominantly observed above clouds. For measurement periods in higher and constant altitudes (calm conditions), which are not shown in the paper, the attitude correction worked properly. For observations below and in clouds, the radiation field is assumed to be dominated by diffuse radiation, which is isotropically distributed. Here, no correction is needed. Based on these limitations and to avoid misinterpretation of the measurements, we decided to show only data measured below and in clouds in Figure 11 and 14. On 14 June 2017 a second cloud layer above the boundary layer cloud guarantied diffuse downward radiation. A discussion on the treatment of the solar radiation in cloud free conditions is added to the revised manuscript (Sect. 3.3.3).

*"Misalignment of the sensor with respect to the horizontal plane affects the measurements of the downward irradiance in cloudless conditions (Wendisch et al. 2001). When the radiation field is dominated by direct solar radiation and the misalignment does not exceed 5-10° (depending on solar zenith angle), a correction is possible as demonstrated by Bannehr and Schwiesow (1993). Due to wind shear, the broadband radiation package started to swing, reaching roll and pitch angles of up to 20° while the heading changed quickly (up to 180° in 5 s). In these conditions, which predominantly occurred above clouds, the algorithm failed to correct the sensor movement. To avoid a misinterpretation of the remaining misalignment bias, data in such conditions had to be excluded from the analysis. In calm wind conditions, the attitude correction worked properly. Below or within clouds it is assumed, that diffuse and isotropically distributed radiation dominates the radiation field. Hence, a correction is not required in these conditions. To identify the presence of clouds shielding direct sunlight, the camera is used."*

- **Page 22, Line 2 & figure 11. The strong negative buoyancy flux just below cloud top might be associated with entrainment. It is hard, however, to find a physical explanation for a strong upward flux just above cloud top – this might be an artefact of calculating an eddy covariance flux from a slant profile that crosses a strong and changing gradient at the inversion where perturbations from the 'mean' can result from changes in the mean rather than true turbulent fluctuations. I would treat this value with extreme caution**
  **and**
  **Page 24, Line 6 – the negative peak in buoyancy flux 'inside the cloud' is located just below the inversion base – again possibly associated with entrainment mixing, though again, calculating a flux from a slant profile across a strong and changing gradient is problematic, and I would want to carefully select the portion of the profile used to stay below the inversion base.**
  Thank you for this comment. We realized that in fact in regions with strong changes in the temperature gradients (e.g. transition from well-mixed to extremely stable regions) the high-pass filter creates artificial variability. Therefore, we now use a Bessel filter (N=20) instead of Savitzky-Golay filter as applied in the discussion paper. This filter has a better performance in regions with a rapid change of the gradient. For moderate inversions with a smoother transition the low-pass filtered signal can now follow the raw signal resulting in reasonable fluctuations. Now, the slant profile variances in Fig. 10, 12 and 13 are of slightly smaller magnitude. Nevertheless, for strong inversions associated with strong changes in temperature gradient we excluded this region from the flux calculation because even the Bessel filter is not able to follow the sharp jump resulting in artificial fluctuations in the region.

p. 15, l. 21: "*On the slant profile, turbulent fluctuations are determined using a high-pass 20th-order Bessel filter. The cut-off frequency $f_c = U/z_c$ for the filter is given by the cloud layer thickness $z_c$ and mean horizontal wind velocity U and is in the range from 0.009 Hz to 0.025 Hz.*"

p. 22, l. 2: Due to the revised filter algorithm, the positive and negative peak is not that strong anymore, but we excluded the temperature inversion region from the slant profile flux calculation, as the filter cannot follow the sharp transition between well-mixed and extremely stable regions.

"*The magnitude of the virtual heat flux derived from the slant profile is significantly smaller than the fluxes measured at constant altitude. Positive heat fluxes of up to 3 W/m$^2$ are measured within the whole mixed layer. The cloud top region is excluded from the virtual heat flux calculation. Due to the strong change in the temperature gradient at the transition from the well-mixed layer to the inversion, the filter algorithm creates artificial fluctuations in this region, which results in unrealistic fluxes.*"

p. 24, l. 6: Due to the revised filter algorithm, the positive and negative peaks are much weaker, but still present. Here, the temperature inversion is not very sharp, therefore no values have to be excluded from the flux calculation.

"*The virtual heat flux fluctuates around zero with a slight negative tendency within the cloud, changing to positive values near the surface. Due to the gradually changing temperature gradient at the inversion, no values have to be excluded from the flux calculation.*"

Furthermore, we calculated the fluxes on the slant profile for overlapping (instead of non-overlapping) 50s-periods to get a more complete picture of their vertical structure.

---

## Author Comment (AC2) · 21 Jun 2019

The authors thank the anonymous referee for the comments, which helped to improve the quality of the manuscript. All suggestions, which were mostly rephrasing issues, were implemented.
* * *

---

## Author Response (AR1)

**Author's response to Ian Brooks' review of: AMT-2019-80**

We thank Ian Brooks for his constructive review, which significantly improved the quality of the manuscript. The author's response is structured as follows:

**Comments of reviewer**

Author's answer to the comment "New text passage"

**General remarks**

- All suggestions for rephrasing and including references were implemented. This remarkably improved the language quality and made the manuscript more comprehensive.
- The term buoyancy flux, which was used throughout the discussion paper, is misleading. For the flux calculation, we used the definition of the sensible heat flux, but with the potential virtual temperature instead of the potential temperature. We renamed *"buoyancy flux HB"* as *"virtual heat flux H"*. This term was introduced by Angevine, 1993 (doi: 10.1175/1520-0450(1993)032<1901:VHFMFA>2.0.CO;2) and the reference is now included in the manuscript.

**Answers to specific reviewer comments**

• Page 4, Line 9: The quoted 'standard lapse rate' of 6.5 k km-1 is rather low, the dry lapse rate is approximately 9.8 K/km (AMS glossary), and at freezing point in just saturated air is 9.968 K/km. The value given here is approximately the wet adiabatic lapse rate. Ideally the wet/dry value should be used in/out of cloud to give most accurate height.

You are right, but here we use the value given for the ICAO standard atmosphere (z

Page 19, Line 33: in discussion of vertical wind, it is worth noting here that these regions of significant upward/downward motion cannot be mean motions, but the sampling of up/down moving portions of a large, boundary-layer scale eddy. It is notable, however, that their vertical location coincides with the regions of higher/lower horizontal wind. Possibly this too is sampling of a large scale eddy? A first idea was that this might result from an artefact in the wind vector transformation. For the balloon, the wind/ turbulence and the motion of the instrument itself are closely connected, which makes the transformation challenging. Therefore, we verified again the transformation of the wind vector during the ascent and could not find, that the vertical wind variation results from a measurement artefact (e.g., an offset in the pitch angle). We are now confident that the shown wind velocity components are correct. Your interpretation of sampling within one large-scale eddy is convincing and in the manuscript we made a short comment about this possible explanation:

"As the location of upward motion and increased horizontal wind coincides, this might be an indication for a large, boundary-layer scale eddy."

A minor issue of phrasing here (and in discussion of standard deviation of w) – the phrase 'wind speed' tends to imply a mean quantity, and thus 'vertical wind speed' seems inappropriate for w, for which simply 'vertical velocity' is perhaps better terminology.

"Wind speed" was replaced by "wind velocity", when it did not refer to the mean value.

• Page 21, Figure 11 caption. Here and elsewhere I think the term 'terrestrial' to refer to the infra red irradiance is misleading, since much of the infra red budget has nothing to do with the surface. 'infra red' would be preferable.

As our analysis focuses mostly on clouds, the term 'terrestrial' might truly be misleading. Thus, we will change the term to "thermal infrared" (TIR). According to the definition of TIR from ~3-50  $\mu$ m (Manfred Wendisch and Ping Yang, 2012), this range will closely agree to our measured values of 4.5-42  $\mu$ m.

• Figure 11 and 14 – why do the solar irradiances stop at cloud top, while the infra red values extend above?

The misalignment of the solar sensors exposed to direct sunlight are usually corrected using the method described by Bannehr and Schwiesow (1993). In case of a strong

movement and misalignment of the sensor package, the correction could not be applied successfully and a bias remained. These conditions with strong wind shear where predominantly observed above clouds. For measurement periods in higher and constant altitudes (calm conditions), which are not shown in the paper, the attitude correction worked properly. For observations below and in clouds, the radiation field is assumed to be dominated by diffuse radiation, which is isotropically distributed. Here, no correction is needed. Based on these limitations and to avoid misinterpretation of the measurements, we decided to show only data measured below and in clouds in Figure 11 and 14. On 14 June 2017 a second cloud layer above the boundary layer cloud guarantied diffuse downward radiation. A discussion on the treatment of the solar radiation in cloud free conditions is added to the revised manuscript (Sect. 3.3.3).

"Misalignment of the sensor with respect to the horizontal plane affects the measurements of the downward irradiance in cloudless conditions (Wendisch et al. 2001). When the radiation field is dominated by direct solar radiation and the misalignment does not exceed 5-10° (depending on solar zenith angle), a correction is possible as demonstrated by Bannehr and Schwiesow (1993). Due to wind shear, the broadband radiation package started to swing, reaching roll and pitch angles of up to 20° while the heading changed quickly (up to 180° in 5 s). In these conditions, which predominantly occurred above clouds, the algorithm failed to correct the sensor movement. To avoid a misinterpretation of the remaining misalignment bias, data in such conditions had to be excluded from the analysis. In calm wind conditions, the attitude correction worked properly. Below or within clouds it is assumed, that diffuse and isotropically distributed radiation dominates the radiation field. Hence, a correction is not required in these conditions. To identify the presence of clouds shielding direct sunlight, the camera is used."

• Page 22, Line 2 & figure 11. The strong negative buoyancy flux just below cloud top might be associated with entrainment. It is hard, however, to find a physical explanation for a strong upward flux just above cloud top – this might be an artefact of calculating an eddy covariance flux from a slant profile that crosses a strong and changing gradient at the inversion where perturbations from the 'mean' can result from changes in the mean rather than true turbulent fluctuations. I would treat this value with extreme caution

**and**

Page 24, Line 6 – the negative peak in buoyancy flux 'inside the cloud' is located just below the inversion base – again possibly associated with entrainment mixing, though again, calculating a flux from a slant profile across a strong and changing gradient is problematic, and I would want to carefully select the portion of the profile used to stay below the inversion base.

Thank you for this comment. We realized that in fact in regions with strong changes in the temperature gradients (e.g. transition from well-mixed to extremely stable regions) the high-pass filter creates artificial variability. Therefore, we now use a Bessel filter (N=20) instead of Savitzky-Golay filter as applied in the discussion paper. This filter has a better performance in regions with a rapid change of the gradient. For moderate inversions with a smoother transition the low-pass filtered signal can now follow the raw signal resulting in reasonable fluctuations. Now, the slant profile variances in Fig. 10, 12 and 13 are of slightly smaller magnitude. Nevertheless, for strong inversions associated with strong changes in temperature gradient we excluded this region from the flux calculation because even the Bessel filter is not able to follow the sharp jump resulting in artificial fluctuations in the region.

p. 15, l. 21: "On the slant profile, turbulent fluctuations are determined using a high-pass 20th-order Bessel filter. The cut-off frequency  $f_c = U/z_c$  for the filter is given by the cloud layer thickness  $z_c$  and mean horizontal wind velocity U and is in the range from 0.009 Hz to 0.025 Hz."

p. 22, l. 2: Due to the revised filter algorithm, the positive and negative peak is not that strong anymore, but we excluded the temperature inversion region from the slant profile flux calculation, as the filter cannot follow the sharp transition between well-mixed and extremely stable regions.

"The magnitude of the virtual heat flux derived from the slant profile is significantly smaller than the fluxes measured at constant altitude. Positive heat fluxes of up to 3 W/m2 are measured within the whole mixed layer. The cloud top region is excluded from the virtual heat flux calculation. Due to the strong change in the temperature gradient at the transition from the well-mixed layer to the inversion, the filter algorithm creates artificial fluctuations in this region, which results in unrealistic fluxes."

p. 24, l. 6: Due to the revised filter algorithm, the positive and negative peaks are much weaker, but still present. Here, the temperature inversion is not very sharp, therefore no values have to be excluded from the flux calculation.

"The virtual heat flux fluctuates around zero with a slight negative tendency within the cloud, changing to positive values near the surface. Due to the gradually changing temperature gradient at the inversion, no values have to be excluded from the flux calculation."

Furthermore, we calculated the fluxes on the slant profile for overlapping (instead of nonoverlapping) 50s-periods to get a more complete picture of their vertical structure.

**Author's response to Anonymous Referee #2**

The authors thank the anonymous referee for the comments, which helped to improve the quality of the manuscript. All suggestions, which were mostly rephrasing issues, were implemented.

**The new BELUGA setup for collocated turbulence and radiation measurements using a tethered balloon: First applications in the cloudy Arctic boundary layer**

Ulrike Egerer1, Matthias Gottschalk2, Holger Siebert1, André Ehrlich2, and Manfred Wendisch2

1Leibniz Institute for Tropospheric Research, Permoserstr. 15, 04318 Leipzig, Germany

2University of Leipzig, Institute for Meteorology, Stephanstr. 3, 04103 Leipzig, Germany

Correspondence: Ulrike Egerer (egerer@tropos.de)

Abstract. The new BELUGA (Balloon-bornE moduLar Utility for profilinG the lower Atmosphere) tethered balloon system is introduced. It combines a set of instruments to measure turbulent and radiative parameters and energy fluxes. BELUGA enables collocated measurements either at a constant altitude or as vertical profiles up to 1.5 km height. In particular, the instrument payload of BELUGA comprises three modular instrument packages for high resolution meteorological, wind vector

- 5 and broadband radiation measurements. The collocated Collocated data acquisition allows to estimate for estimates of the driving parameters of in the energy balance in various altitudes 
[revised manuscript text omitted]

(a) Tethered balloon system Ultrasonic Transducers Inertial measurement unit TROPOS Temperature and humidity sensors Battery and data aquisition 1.5 m © S Schön Sächsische Zeitung (c) Hotwire anemometer package (HP) (d) Broadband radiation package (BP) Hot-wire Temperature and humidity Thermal infrared sensor Solar irradiance sensors (up- and downward) irradiance (up- and downward)

Battery and

data aquisition

Figure 1. The tethered balloon system on the ice floe next to RV Polarstern (a) and photographs of the individual instrument packages for turbulence (b and c) and radiation (d).

allows a larger mass of the payload. It includes the UP for turbulence and one BP for radiation measurements. For strong wind conditions, the payload is reduced to reach a sufficient maximum altitude. This configuration 2 comprises the HP and one BP. Configuration 3 is applied in most favorable conditions (low and uniform wind speed), comprising of the UP and two BPs. A separate standard meteorology package routinely measures air temperature, relative humidity, wind speed-velocity and altitude,

and transmits the data to ground for online monitoring. The ultrasonic anemometer, with a weight of around 6 kg, is attached 5 at a fixed point in a distance of 20 m below the balloon; the other more lightweight payloads HW and BP (

**Figure 2.** Three main instrument configurations on the tethered balloon: (1) configuration 1 with an ultrasonic anemometer package (UP) and a broadband radiation package (BP), (2) configuration 2 with a BP and a hot-wire anemometer package (HWHP) and (3) configuration 3 with an UP and two BPs. Due to the modular approach, further configurations are possible. Distances and dimensions are not to scale.

with the standard adiabatic lapse rate  $L_0 = 6.5 \text{ K km}^{-1}$  and the gas constant for dry air  $R = 287 \text{ J kg K}^{-1}$ . The ground (Wendisch and Brenguier, 2013). The surface temperature  $T_0$  is measured at a nearby meteorological mast. The ground surface pressure  $p_0$  is a 10 second average of the payload's  $p_b$  before the start of the flight. The barometric pressure is corrected for the pressure change over each flight. This standardized procedure ensures comparable altitudes.

**5 2.2 Measurement strategy**

The sampling strategy is based on two different approaches: (1) Keeping BELUGA at a constant altitude for a time period of typically 10–15 minutes. In this case, the data provide a statistical basis for turbulent flux estimates or to characterize the time evolution of the radiative cloud properties. (2) A continuous ascent or descent through the ABL yields a vertical profile to study the vertical distribution of ABL parameters.

- 10 Measurements close to the ground surface are used for a comparison to the 10 m high meteorological mast. Figure 3 shows an exemplary height record for one complete flight, which includes all elements of the measurement strategy. After an hour-long measurement period at groundnear the surface, a continuous ascent is performed. This provides an overview of the boundary layer using the online measurements of the meteorology instrument package and is the basis for the measurements of the second part of the flight: Levels for continuous measurements at fixed altitudes were defined around the temperature inversion
- 15 and inside the cloud layer.